# Deep learning-based feature extraction for prediction and interpretation of sharp-wave ripples in the rodent hippocampus

Andrea Navas-Olive[1†‡], Rodrigo Amaducci[2†‡], Maria-Teresa Jurado-Parras[1], Enrique R Sebastian[1], Liset M de la Prida[1*]

[1]Instituto Cajal, CSIC, Madrid, Spain; [2]Grupo de Neurocomputación Biológica (GNB), Universidad Autónoma de Madrid, Madrid, Spain

*For correspondence: lmprida@cajal.csic.es

†These authors contributed equally to this work

‡Co-shared first author

**Abstract** Local field potential (LFP) deflections and oscillations define hippocampal sharp-wave ripples (SWRs), one of the most synchronous events of the brain. SWRs reflect firing and synaptic current sequences emerging from cognitively relevant neuronal ensembles. While spectral analysis have permitted advances, the surge of ultra-dense recordings now call for new automatic detection strategies. Here, we show how one-dimensional convolutional networks operating over high-density LFP hippocampal recordings allowed for automatic identification of SWR from the rodent hippocampus. When applied without retraining to new datasets and ultra-dense hippocampus-wide recordings, we discovered physiologically relevant processes associated to the emergence of SWR, prompting for novel classification criteria. To gain interpretability, we developed a method to interrogate the operation of the artificial network. We found it relied in feature-based specialization, which permit identification of spatially segregated oscillations and deflections, as well as synchronous population firing typical of replay. Thus, using deep learning-based approaches may change the current heuristic for a better mechanistic interpretation of these relevant neurophysiological events.

## Editor's evaluation

This paper will be of interest to the neuroscience community studying brain oscillations. It presents a new method to detect sharp wave-ripples in the hippocampus with deep learning techniques, instead of the more traditional signal processing approach. The overall detection performance improves and this technique may help in identifying and characterizing previously undetected physiological events.

## Introduction

Interpreting brain signals is essential in understand cognition and behavior. Biologically relevant oscillations are considered reliable markers of brain operation (*Buzsáki et al., 2012*; *Friston et al., 2015*). Thus, analysis of either surface electroencephalography (EEG) or intracranial local field potential (LFP) is typically based on spectral methods relying on gold-standard definitions (*Niedermeyer and Silva, 2005*). However, other features of EEG/LFP signals such as the slope, polarity, and latency to events are equally important (*Modi and Sahin, 2017*). While interpreting neurophysiological signals is strongly influenced by this heuristics, methodological issues limit further advances.

**eLife digest** Artificial intelligence is finding greater use in society through its ability to process data in new ways. One particularly useful approach known as convolutional neural networks is typically used for image analysis, such as face recognition. This type of artificial intelligence could help neuroscientists analyze data produced by new technologies that record brain activity with higher resolution.

Advanced processing could potentially identify events in the brain in real-time. For example, signals called sharp-wave ripples are produced by the hippocampus, a brain region involved in forming memories. Detecting and interacting with these events as they are happening would permit a better understanding of how memory works. However, these signals can vary in form, so it is necessary to detect several distinguishing features to recognize them.

To achieve this, Navas-Olive, Amaducci et al. trained convolutional neural networks using signals from electrodes placed in a region of the mouse hippocampus that had already been analyzed, and 'telling' the neural networks whether they got their identifications right or wrong. Once the networks learned to identify sharp-wave ripples from this data, they could then apply this knowledge to analyze other recordings. These included datasets from another part of the mouse hippocampus, the rat brain, and ultra-dense probes that simultaneously assess different brain regions. The convolutional networks were able to recognize sharp-wave ripple events across these diverse circumstances by identifying unique characteristics in the shapes of the waves.

These results will benefit neuroscientists by providing new tools to explore brain signals. For instance, this could allow them to analyze the activity of the hippocampus in real-time and potentially discover new aspects of the processes behind forming memories.

During memory consolidation and retrieval, the hippocampal system releases short memory traces in the form of neuronal sequences (*Joo and Frank, 2018*; *Pfeiffer, 2020*; *Pfeiffer and Foster, 2015*). Such activity comes often in tandem with spatially segregated oscillations (100–250 Hz) and LFP deflections dubbed sharp-wave ripples (SWRs) (*Buzsáki, 2015*). They result from active recruitment of dedicated cell-type-specific microcircuits (*de la Prida, 2020*; *Stark et al., 2014*; *Valero et al., 2015*). SWR-associated sequences can either replay previous experience or preplay internal representations (*Farooq and Dragoi, 2019*; *Foster, 2017*; *Joo and Frank, 2018*), making their automatic detection crucial in understanding memory function. However, while spectral-based filters have permitted real-time SWR-related interventions (*Fernández-Ruiz et al., 2019*; *Girardeau et al., 2009*; *Jadhav et al., 2012*), these methods are not optimal to disambiguate the underlying variability of a wealth of events, especially during online operation. Moreover, with the advent of ultra-dense recordings, the need for automatic identification is pressing. In spite of recent advances (*Dutta et al., 2019*; *Hagen et al., 2021*), current solutions still require improvement to capture the complexity of SWR events across hippocampal layers.

Here, we exploit the extraordinary capability of convolutional neural networks (CNNs) for real-time recognition to identify SWR (*Bai et al., 2018*). Instead of adopting standard approaches used for temporal data such as in speech recognition, we chose to rely on unfiltered LFP profiles across hippocampal strata as individual data points making up an image. The one-dimensional object is equivalent to a clip of one-row pixels with as many colors as LFP channels. We show how one-dimensional CNN operating over high-density LFP hippocampal signals overcome spectral methods in detecting a large variety of SWR. Moreover, we develop a strategy to decode and explain CNN operation. In doing so, we discovered some features of SWR that permit their detection at distant layers when applied to Neuropixels recordings (*Jun et al., 2017*). Using these tools allow for a more comprehensive interpretation of SWR signatures across the entire hippocampal system.

## Results
### Artificial neural network architecture and operation
Inspired by You-Only-Look-Once (YOLO) networks for real-time object recognition (*Redmon et al., 2015*), we adapted a CNN architecture to search for SWR in the dorsal hippocampus of awake head-fixed mice. LFP signals acquired with high-density 8-channel silicon probes provide detailed

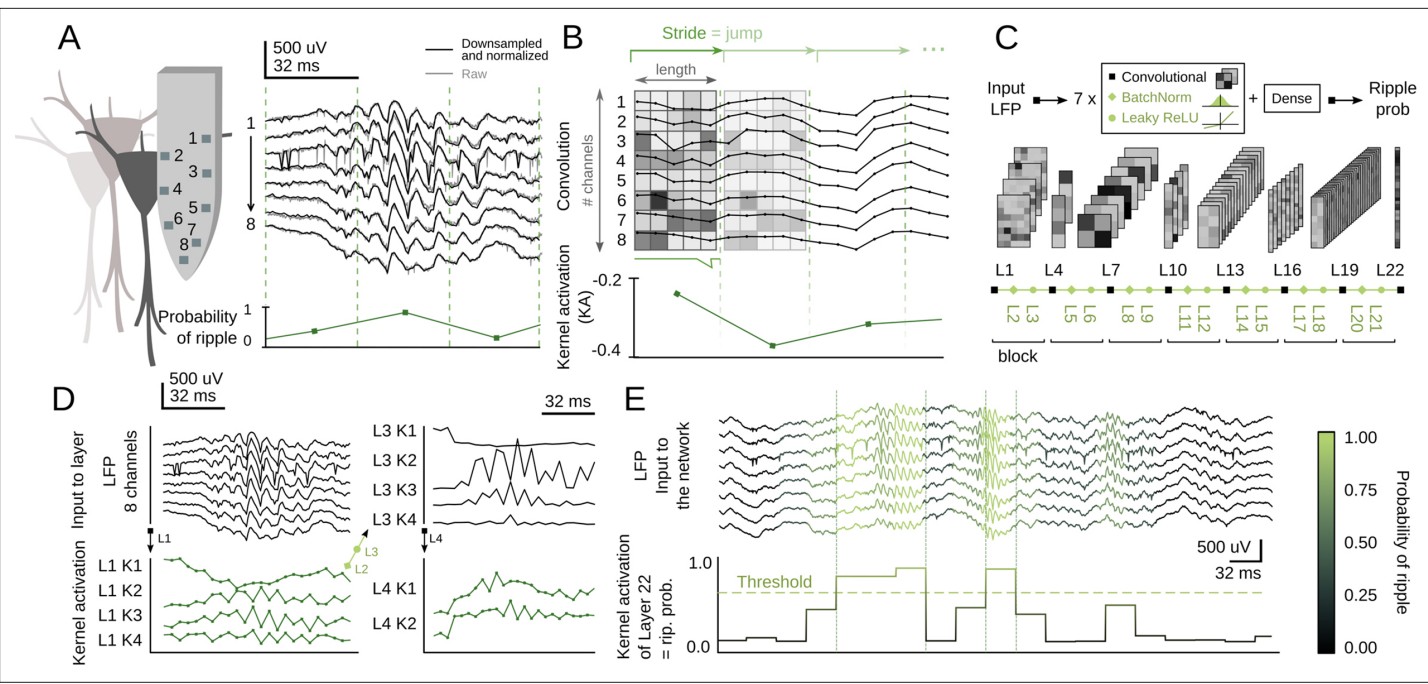

**Figure 1.** Convolutional neural network (CNN) definition and operation. (**A**) Example of a sharp-wave ripple (SWR) event recorded with 8-channel silicon probes in the dorsal CA1 hippocampus of head-fixed awake mice. Vertical lines mark the analysis window (32 ms). The probability of SWR event from each window is shown at bottom. (**B**) Example of L1 kernel operation and calculation of the kernel activation (KA) signal. (**C**) Network architecture consists of seven blocks of one Convolutional layer+one BatchNorm layer+one Leaky ReLU layer each (layers 1–21). Dense layer 22 provides the CNN output as the SWR probability. (**D**) Examples of KA for layers 1–4 resulting from the SWR event shown in A. Note how the 8-channel local field potential (LFP) input is progressively transformed to capture different features of the event. (**E**) Example of the CNN output (i.e. KA of layer 22) at 32 ms resolution. A probability threshold can be used to identify SWR events. Note that some events can be predicted well in advance.

The online version of this article includes the following figure supplement(s) for figure 1:

**Figure supplement 1.** Network definition and parameters.

information about the underlying CA1 microcircuit (*Figure 1A*; *Mizuseki et al., 2011*; *Navas-Olive et al., 2020*). The goal of the artificial network operating over 8-channel input signals (down-sampled at 1250 Hz) was to provide a single-output probability for the occurrence of an SWR event in a given temporal window (*Figure 1A*, bottom trace). Therefore, the input 'object' is equivalent to a stream of pixels (×1 number of data samples) with 8-channels instead of colors.

Convolutional layers search for particular features in the input data by using kernels. The kernels of the first layer (L1) have dimensions of 8-channels × length, with length reflecting the number of data samples. They advance along the temporal axis moving forward a similar number of non-overlapping samples defined by the stride (*Figure 1B*). The result of this operation is the kernel activation (KA) signal, which reflects the presence of some input features. L1 kernel length should be defined by considering the desired output resolution of the network. To ease subsequent online applications, we chose either 32 ms (CNN32, L1 kernel length 5) or 12.8 ms resolution (CNN12, L1 kernel length 2).

Our CNN operates by receiving the 8-channels input into each of the four kernels of L1 (*Figure 1C*). Kernels process the LFP and output a KA signal (*Figure 1D*). After passing through L1, the 8-channels are transformed into 4-channels, one per kernel (e.g. L1K1, L1K2, etc.). L1 output is then transformed by a BatchNorm layer (L2) and a Leaky ReLU layer (L3), before entering the next block (L4-L5-L6 and so on; *Figure 1C*). The size of subsequent kernels is defined by the input data from the Convolutional layers of the previous block (see Materials and methods). Inspired by YOLO, we staggered blocks with kernels of large and short length to allow for alternate convolution of the temporal and channel axes. As data are processed along these blocks, resolution decreases and hence the kernel length becomes progressively shorter.

We defined a suitable number of blocks that optimized the input (8 channels) and output features (1 channel output at 32 ms or 12.8 resolution), resulting in seven blocks for a total of 21 layers

(*Figure 1C*). The final layer (L22) is a Dense layer with a sigmoidal activation function, so that the CNN output (between 0 and 1) can be interpreted as the SWR probability. An SWR event can be detected using an adjustable probability threshold (*Figure 1E*). Note that our CNN network operates along all streamed LFP data without any specification of the ongoing oscillatory state (i.e. theta or non-theta segments accompanying running and immobility periods, respectively).

## CNN training and performance offline and online

Having defined the main network architecture, we used a dataset manually tagged by an expert for training and initial validation (1794 events, two sessions from two mice; *Supplementary file 1*). An important decision we made was manually annotating the start and the end of SWR events so that the CNN could learn their onset.

Given the large number of parameter combinations, we run two optimization rounds using training and test chunks from the training dataset. We first tested a subset of hyper-parameters to look for the 10-best networks (*Figure 1—figure supplement 1A*, green shaded), and chose the one with the lowest and more stable learning curve (*Figure 1—figure supplement 1B*, arrowhead). Stabilization of the loss function error for the training and test subsets along epochs excluded potential overfitting (*Figure 1—figure supplement 1C*). In order to compare CNN performance against spectral methods, we implemented a Butterworth filter, which parameters were optimized using the same training set (*Figure 1—figure supplement 1D*). A subsequent hyper-parameter search (781 combinations)

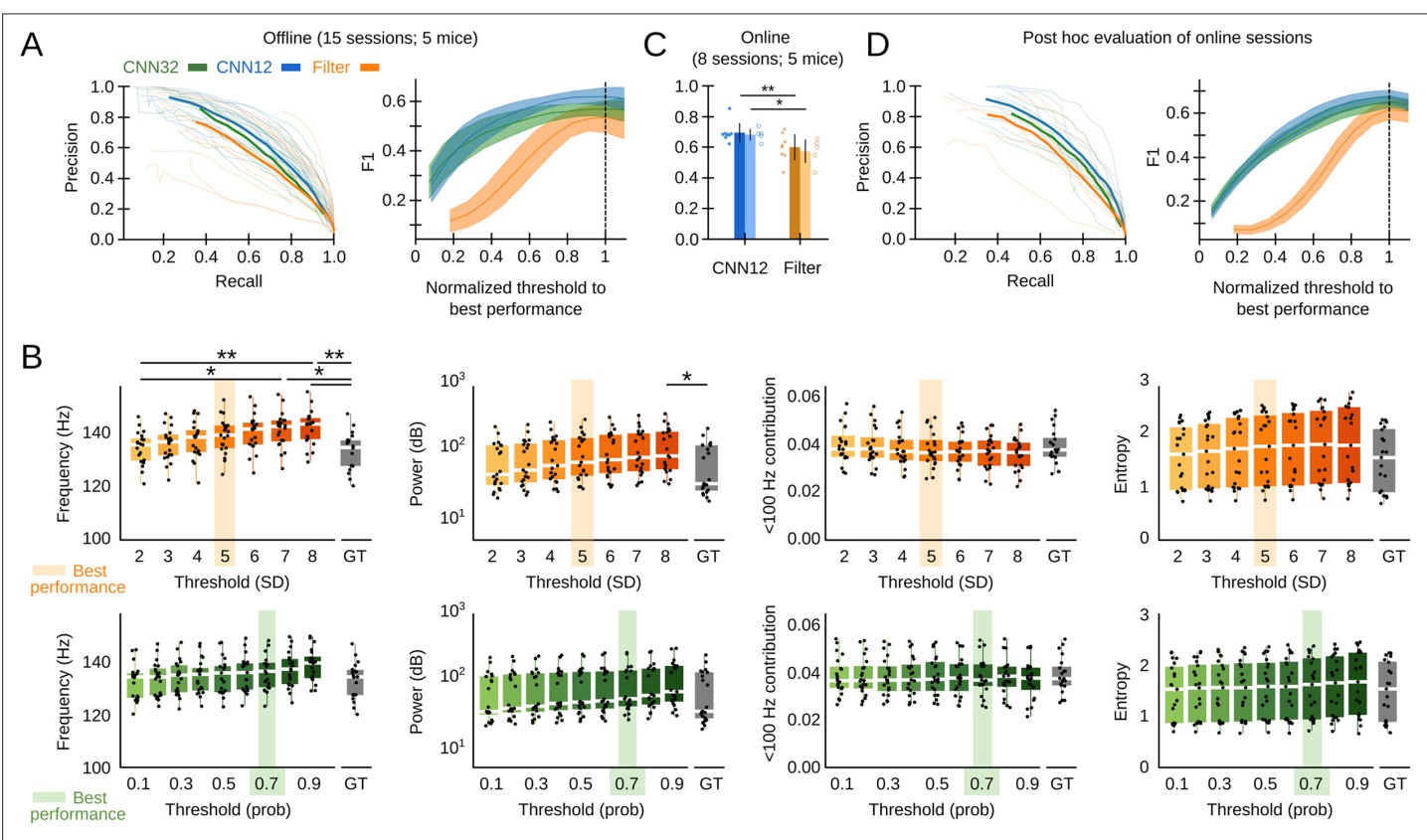

**Figure 2.** Convolutional neural network (CNN) performance. (**A**) Offline P-R curve (mean is dark; sessions are light) (left), and F1 score as a function of normalized thresholds for the CNN at 32 and 12.8 ms resolution as compared with the Butterworth filter (right). Data reported as mean±95% confidence interval for validation sessions (n=15 sessions; five mice). (**B**) Comparison of mean sharp-wave ripple (SWR) features (frequency, power, high-frequency band contribution, and spectral entropy) of events detected offline by the filter (upper plots) and the CNN32 (bottom) as a function of the threshold. The mean best threshold is indicated (5SD for the filter, 0.7 probability for the CNN). Note effect of the threshold in the mean frequency value (Kruskal-Wallis, Chi2(7)=30.5, p<0.0001; post hoc tests *, p<0.05; **, p<0.001) and the power (Kruskal-Wallis, Chi2(7)=16.4, p=0.0218) for the filter but not for the CNN. Note also, differences against the mean value in the ground truth (GT). Mean data from n=15 sessions; five mice. (**C**) Online detection performance of CNN12 as compared with the Butterworth filter (n=8 sessions, t-test p=0.0047; n=5 mice, t-test p=0.033). (**D**) Mean and per session P-R curve (left), and F1 score as a function of the optimized threshold for online sessions, as analyzed post hoc (right). Data from n=8 sessions from five mice.

confirmed that the trained CNN was in the top-30 group (*Figure 1—figure supplement 1E*). A code notebook is available at https://colab.research.google.com/github/PridaLab/cnn-ripple/blob/main/src/notebooks/cnn-example.ipynb. The trained model is accessible at the GitHub repository both for Python: https://github.com/PridaLab/cnn-ripple (copy archived at swh:1:rev:9dcc5b6a8267b89e-b86a2813dbbcb74a621a701b; *Amaducci and Navas-Olive, 2021*) and MATLAB: https://github.com/PridaLab/cnn-matlab; (copy archived at swh:1:rev:060b2ff6e4b6c5eacb9799addd5123ad06eaaf33; *Navas-Olive and Esparza, 2022*).

We assessed the offline performance of the chosen CNN, as compared to the Butterworth filter as the gold standard, using additional tagged sessions never used for training (5695 events from n=15 sessions from five mice; *Supplementary file 1*). Performance was evaluated by calculating the precision (P, proportion of correct predictions over all predictions), recall (R, proportion of correct predictions over ground truth events, also known as sensitivity), and F1 values (harmonic mean of precision and recall). The P-R curve depicted better offline operation of both the CNN12 and CNN32 as compared with the filter (*Figure 2A*, left). To make the CNN and the filter thresholds comparable, we normalized their values by the best threshold performance (0.7 probability threshold for the CNN, 5SD for the filter). When we considered the relationship between performance and the detection threshold, we found that the CNN was more robust than the filter (*Figure 2A*, right). Filter thresholds had effect in biasing detection of SWR, which exhibited different mean feature values (frequency and power) (*Figure 2B*, upper plots). In contrasts, mean features of SWR detected by the CNN did not depend on the threshold and were consistent with the ground truth (*Figure 2B*, bottom).

The offline analysis presented above was possible because the ground truth was already known. In real case scenarios, the experimenter has to rely in relatively arbitrary threshold settings. To evaluate this further, we performed a new set of experiments for real-time detection in the Open Ephys (OE) environment (*Siegle et al., 2017*) (eight sessions from five mice). To this purpose, we developed a plugin designed to incorporate TensorFlow, an open-source library for machine learning applications, into the OE graphic user interface (*Figure 1—figure supplement 1F, G*; *Supplementary file 1*). To be consistent with detection standards (*Fernández-Ruiz et al., 2019*), the online filter was applied to the channel with maximal ripple power and an additional non-ripple channel was used to veto detection of common artifacts. We found better online performance of the CNN at 12.8 ms resolution as compared with the filter (*Figure 2C*; per session p=0.0047; per mice p=0.033). When it came to the ability to anticipate SWR events online, the CNN slightly overtook the Butterworth filter (time-to-SWR-peak for CNN12: –7.01±2.05 ms; Butterworth filter: –4.66±2.87 ms; paired t-test, p=0.048). A post hoc offline evaluation of online sessions confirmed better performance of the CNN versus the filter, for all normalized thresholds (*Figure 2D*).

## Detection limits of SWR and their influences on CNN operation

Are there any practical detection limit for SWR? How good is CNN performance and how much is it determined by the expert heuristics?

First, we sought to compare CNN and the filter at its maximal capability using data from all validation sessions (offline and online: 22 sessions from 10 mice). To this purpose, we equated the methods using the best possible detection threshold per session (the one that optimized F1) and found roughly similar values (*Figure 3A*; CNN12: F1=0.68 ± 0.06; CNN32: F1=0.63 ± 0.05; Butterworth filter: F1=0.65 ± 0.11), indicating the CNN meet the gold standard provided the filter is optimized. Note that this can only be possible because we know the ground truth. Remarkably, the filter exhibited larger variability across sessions. Our CNN performed similar to a filter-based optimized algorithm (F1=0.65 ± 0.11) (*Dutta et al., 2019*), but significantly better than RippleNET, a recurrent network designed to detect SWR mostly during periods of immobility (F1=0.31 ± 0.22; p<0.00001 one-way ANOVA for comparisons with both CNN12 and CNN32) (*Hagen et al., 2021*). This supports similar operation of CNN as compared with the gold standard in conditions when optimized detection is possible (i.e. when the ground truth is known).

The use of supervised learning for training and posterior validation requires using datasets annotated by experts. However, the expert's opinion may be influenced by the recording method, the experimental goal, and the existing knowledge. To evaluate the impact of these potential biases, we used the ground truth from a second expert in the lab for validation purposes only (3403 events, n=14 sessions, seven mice). While results were overall comparable, there were some natural differences

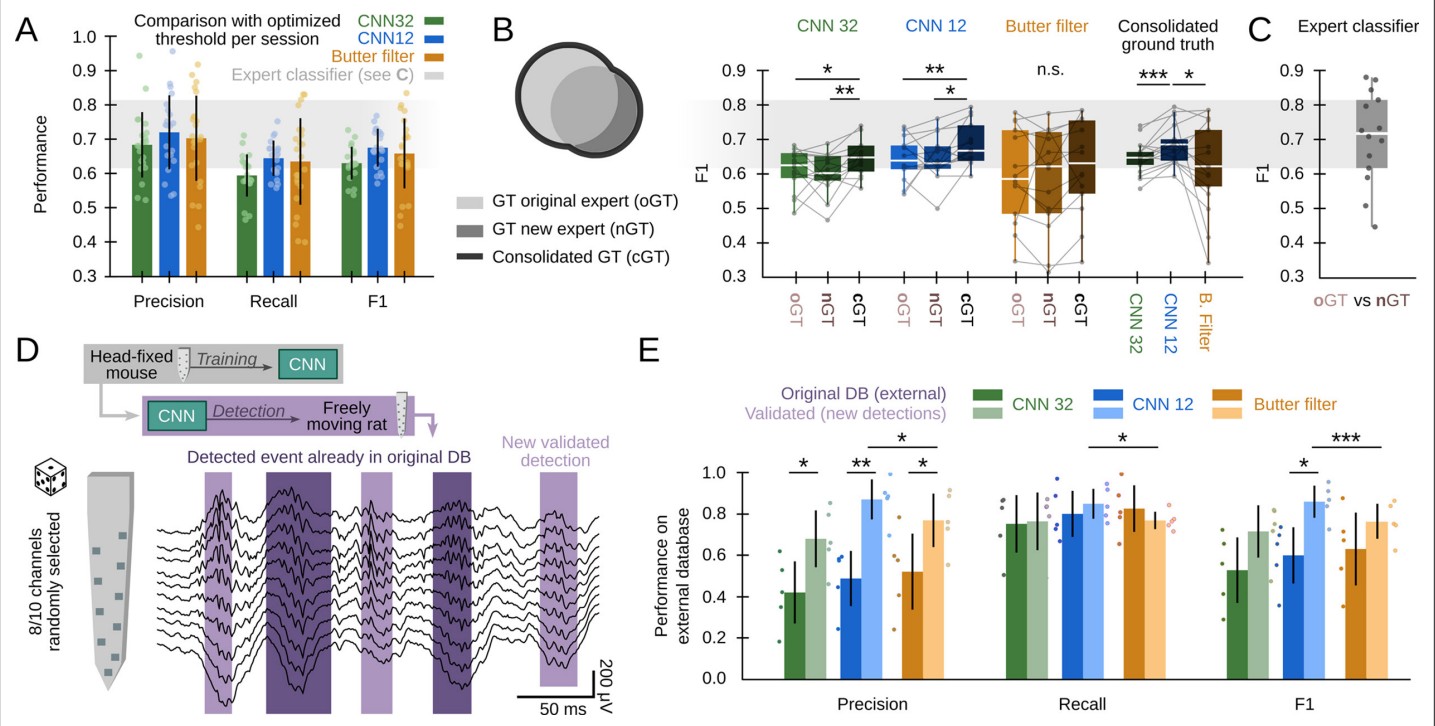

**Figure 3.** Effects of different experts' ground truth on convolutional neural network (CNN) performance. (**A**) Comparison between the CNN and Butterworth filter using thresholds that optimized F1 per session (22 recordings sessions from 10 mice). Note that this optimization process can only be implemented when the ground truth (GT) is known. (**B**) A subset of data annotated independently by two experts was used to evaluate the ability of each method to identify events beyond the individual ground truth. The original expert provided data for training and validation of the CNN. The new expert tagged events independently in a subset of sessions (14 sessions from seven mice). The performance of CNN, but not that of the filter, was significantly better when confronted with the consolidated ground truth (one-way ANOVA for the type of ground truth for CNN32 F(2)=0.01, p=0.0128 and CNN12 F(2)=0.01, p=0.0257). Significant effect of methods when applied to the consolidated ground truth (one-way ANOVA F(2)=0.02, p=0.0331; rightmost); post hoc tests **, p<0.01; ***, p<0.005. CNN models and the filter were applied at mean best performance threshold. (**C**) Performance obtained from the experts' ground truth when acting as a mutual classifier (n=14 sessions). Note that this provides an estimation of the maximal performance level. (**D**) We used the hc-11 dataset (*Grosmark and Buzsáki, 2016*) at the CRCNS public repository (https://crcns.org/data-sets/hc/hc-11/about-hc-11) to further evaluate the effect of the definition of the ground truth and to test for the CNN generalization capability. The data consisted in 10-channel high-density recordings from the CA1 region of freely moving rats. We randomly selected 8-channels to cope with inputs dimension of our CNN, which was not retrained. The dataset comes with annotated sharp-wave ripple (SWR) events (dark shadow) defined by stringent criteria (coincidence of both population synchrony and SWR). CNN False Positives defined by this partially annotated ground truth were re-reviewed and validated (light shadow). (**E**) Performance of the original CNN, without retraining, at both temporal resolutions over the originally annotated (dark colors) and after False Positives validation (light colors). Performance of the Butterworth filter is also shown. Paired t-test at *, p<0.05; **, p<0.001; ***, p<0.001. Data from five sessions, two rats. See *Supplementary file 1*.

between experts in a session-by-session basis (*Figure 3B*). Interestingly, when we confronted the network detection with the consolidated ground truth, we noted that the CNN could be actually detecting many more SWR events than initially accounted by each individual expert (one-way ANOVA for ground truth, CNN12: F(2)=0.01, p=0.026; CNN32: F(2)=0.01, p=0.013). In contrast, the filter failed to exhibit such an improvement, and performed worse when tested against the consolidated ground truth (one-way ANOVA for models, F(2)=0.02, p=0.033) (*Figure 3B*, rightmost). Notably, an expert acting as a classifier of the other expert's ground truth scored at 0.70±0.13 (*Figure 3C*), providing mean reference of best performance (*Figure 3A and B*).

To evaluate this point further, and to test for the capability of the CNN to generalize beyond training with head-fixed mice data, we used an externally annotated dataset of SWR recorded with high-density silicon probes from freely moving rats (*Grosmark and Buzsáki, 2016*; *Figure 3D*; 2041 events; five sessions from two rats; *Supplementary file 1*). In that work, SWR detection was conditioned on the coincidence of both population synchrony and LFP definition, thus providing a 'partial ground truth' (i.e. SWR without population firing were not annotated in the dataset). Consistently, the network recalled most of the annotated events (R=0.80 ± 0.18), but precision was apparently low

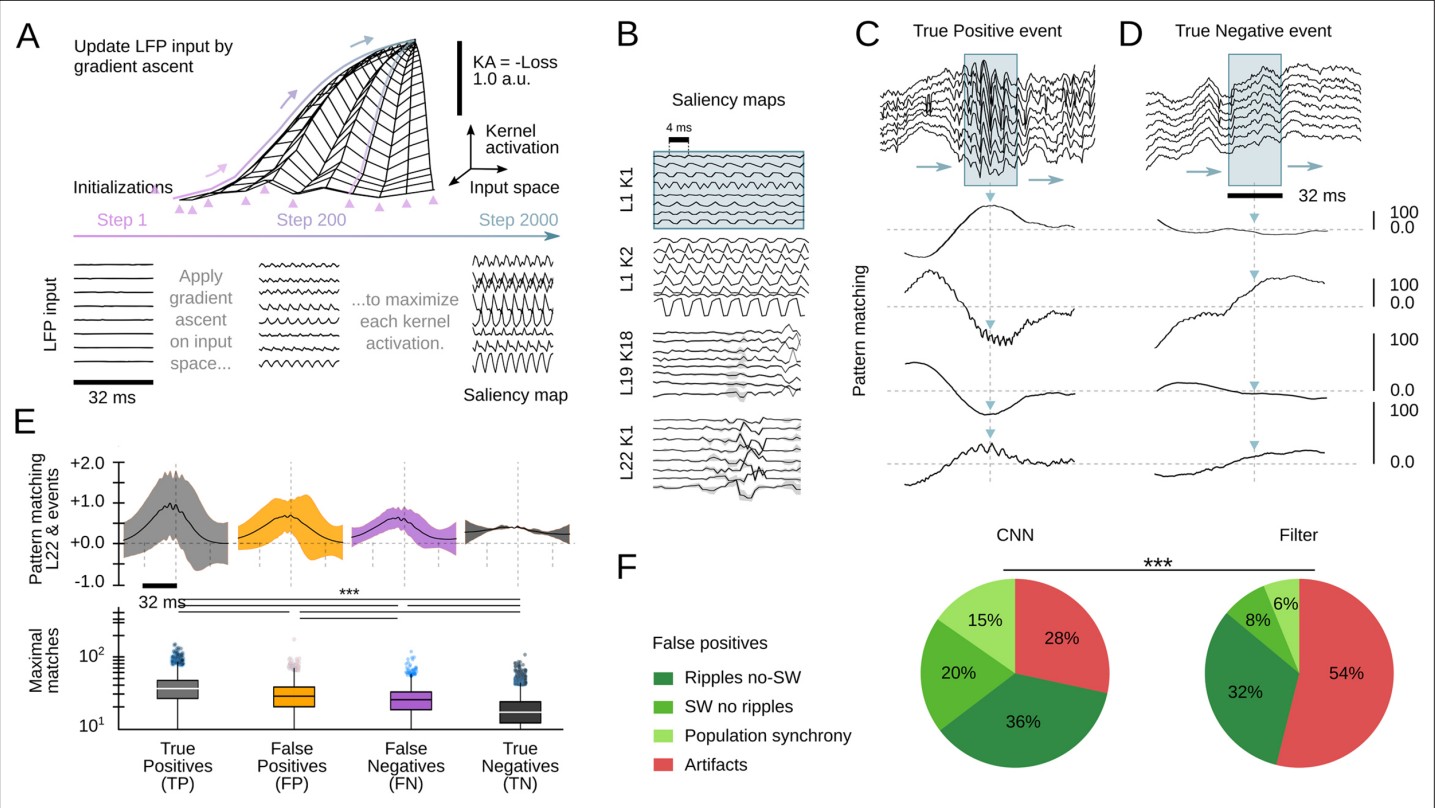

**Figure 4.** Analysis of the convolutional neural network (CNN) kernel saliency maps. (**A**) Schematic illustration of the method to calculate the kernel saliency maps using gradient ascent. Note that different initializations converge to the same solution. (**B**) Examples of saliency maps from some representative kernels. Note ripple-like preferred features of L1 kernels and temporally specific features of L19 and L22 kernels. (**C**) Pattern-matching between saliency maps shown in B and local field potential (LFP) inputs of the example SWR event (120 ms window). (**D**) Same as in C for a True Negative example event. (**E**) Mean template-matching signal (top) and maximal values (bottom) from all detected events classified by CNN32 as True Positive (4385 events), False Positives (2468 events), False Negatives (3055 events), and True Negatives (4902 events). One-way ANOVA, F(3)=1517, p<0.0001; ***, p<0.001 after correction by Bonferroni. (**F**) Distribution of False Positive events per categories both in the CNN32 and the filter.

(P=0.42 ± 0.18) (*Figure 3E*). Hence, we evaluated all False Positive predictions and found that many of them were actually unannotated SWR (2403 events), meaning that precision was actually higher (P=0.77 ± 0.08 for CNN32, P=0.86 ± 0.08; for CNN12, both at P<0.01 for paired t-test; *Figure 3E*). As above, the filter failed to improve F1 performance (*Figure 3E*), and remained lower than for the CNN12.

Altogether, our analyses indicate that detection limits of SWR may be determined by the expert's criteria. CNN performance improves when confronted with the consolidated ground truth, supporting that shared community tagging may help to advance our understanding of SWR definition. Importantly, a CNN trained in data from head-fixed mice was able to generalize to freely moving rats.

## Unveiling SWR latent features

Interpretability is a major issue in modern machine learning (*Mahendran and Vedaldi, 2014*; *Richards et al., 2019*). To better understand and validate CNN operation, we looked for methods to visualize the kernel features that had better explained the network ability to recognize SWR events. We exploited a standard procedure from CNN image recognition (*Simonyan et al., 2013*) consisting on maximizing the KA using gradient ascent in the input space (*Figure 4A*, top). To this purpose, a noisy LFP input is progressively updated until the KA is maximal, using different initialization values (*Figure 4A*, bottom). The resulting signal is equivalent to a saliency map reflecting the latent preferred features by each CNN kernel. This approach is similar to infer visual receptive fields using noise stimulation.

Similar as two-dimensional CNN layers specialize in detecting edges, shapes, and textures of an image, we found the kernels focused in distinct LFP features. Consistently with data above, kernels

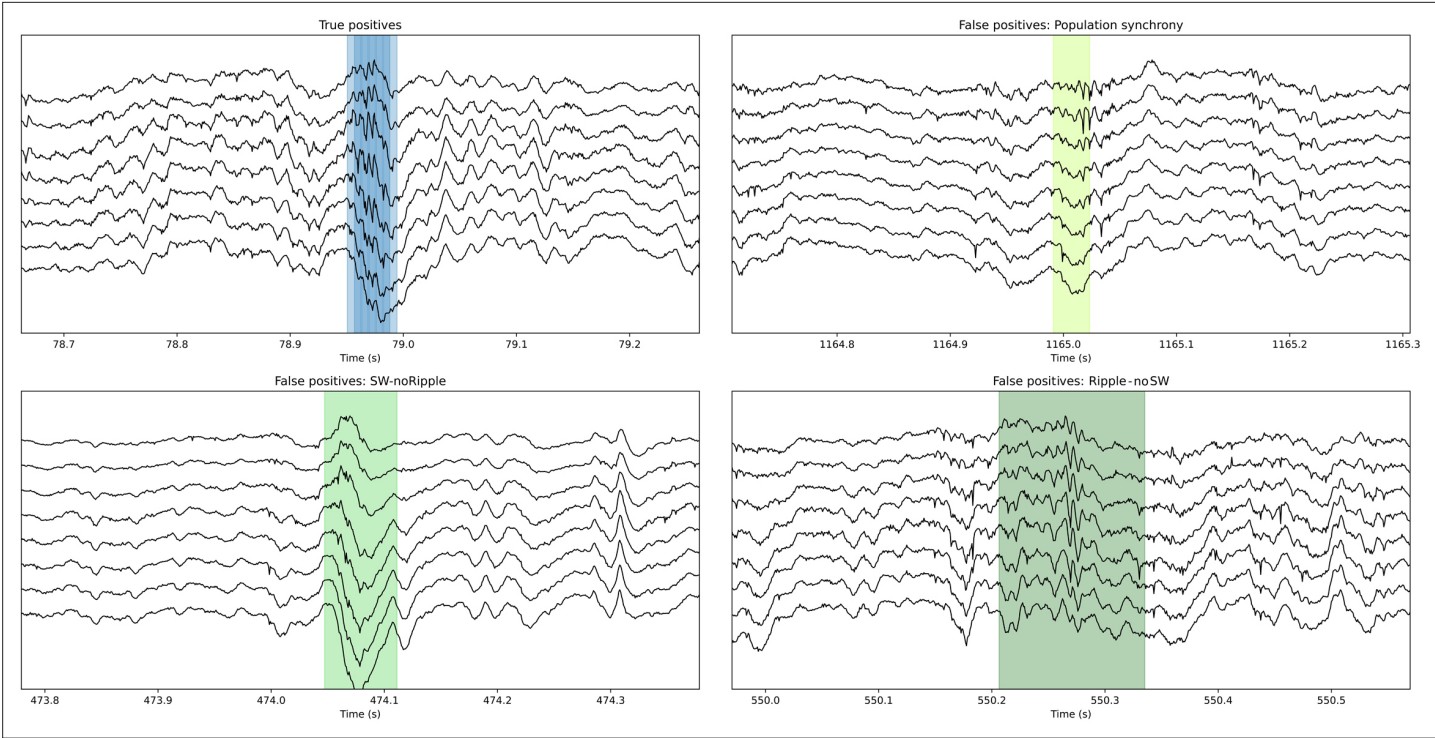

**Figure 5.** Examples of True Positive and False Positive detections by the convolutional neural network (CNN). Note that some False Positive events are sharp waves without ripples (SW no ripple) and sharp wave with population firing. The CNN also detected ripples with no clear associated sharp wave (ripple no SW). While all these False Positive types of events are not included in the ground truth, they resemble physiological relevant categories. This figure is built with an executable code: https://colab.research.google.com/github/PridaLab/cnn-ripple-executable-figure/blob/main/cnn-ripple-false-positive-examples.ipynb.

from the first layers specialized in detecting rhythmic and periodic patterns (e.g. L1K1 and L1K2), while later layers seem to focus in identifying these patterns along time (e.g. L19K18; *Figure 4B*). By computing the pattern-matching function between saliency maps and the 8-channels LFP, we evaluated how the kernels accounted for different features of True Positive events, that is, SWR (*Figure 4C*). For example, L1K1 was maximally activated at the peak of ripple oscillations, while L1K2 and L19K18 were maximal at the onset, supporting the network ability to anticipate SWR. Pattern-matching between true SWR events and the saliency map of the output layer L22 provided an idea of what the CNN recognized as an ideal 'object'. In contrast, pattern-matching values in the absence of SWR events (True Negative events) were typically lower as compared with those obtained from the ground truth (*Figure 4D*).

To quantify these observations, we evaluated how much the output of L22K1 saliency maps matched different input events, using data from the training and offline validation sessions (17 sessions, seven mice). Consistent with the examples, pattern-matching was maximal for True Positive and minimal for True Negative events (one-way ANOVA, $F(3)=1517$, $p<0.0001$). Pattern-matching values were higher for False Positives than for False Negatives (*Figure 4E*), meaning that the network may be identifying some latent features. A closer examination of False Positive predictions suggested that about 20% of them could be reclassified. From these, about one-third were sharp waves without clear associated ripples (SW no ripples), while others were actually ripples events without associated sharp waves (ripples no SW), population firing, and artifacts (*Figure 4F*). Instead, examination of False Positive by the filter showed a major trend to detect artifacts at the expenses of more physiologically relevant events (*Figure 4F*). Examples of True Positive and False Positive detected by the CNN can be seen in *Figure 5*.

This analysis confirms that the CNN has the ability to identify SWR events by relying on feature-based kernel operation. Moreover, some ambiguous predictions according to the current definition of SWR may identify different forms of population firing and oscillatory activities associated to sharp waves, supporting the network ability to generalize beyond the particular expert's ground truth.

## Interpreting and explaining CNN operation

As shown above, the CNN ability relies on feature extraction by the different kernels. To gain explanatory power on how this applies to SWR detection, we sought to visualize and quantify the CNN kernel operation.

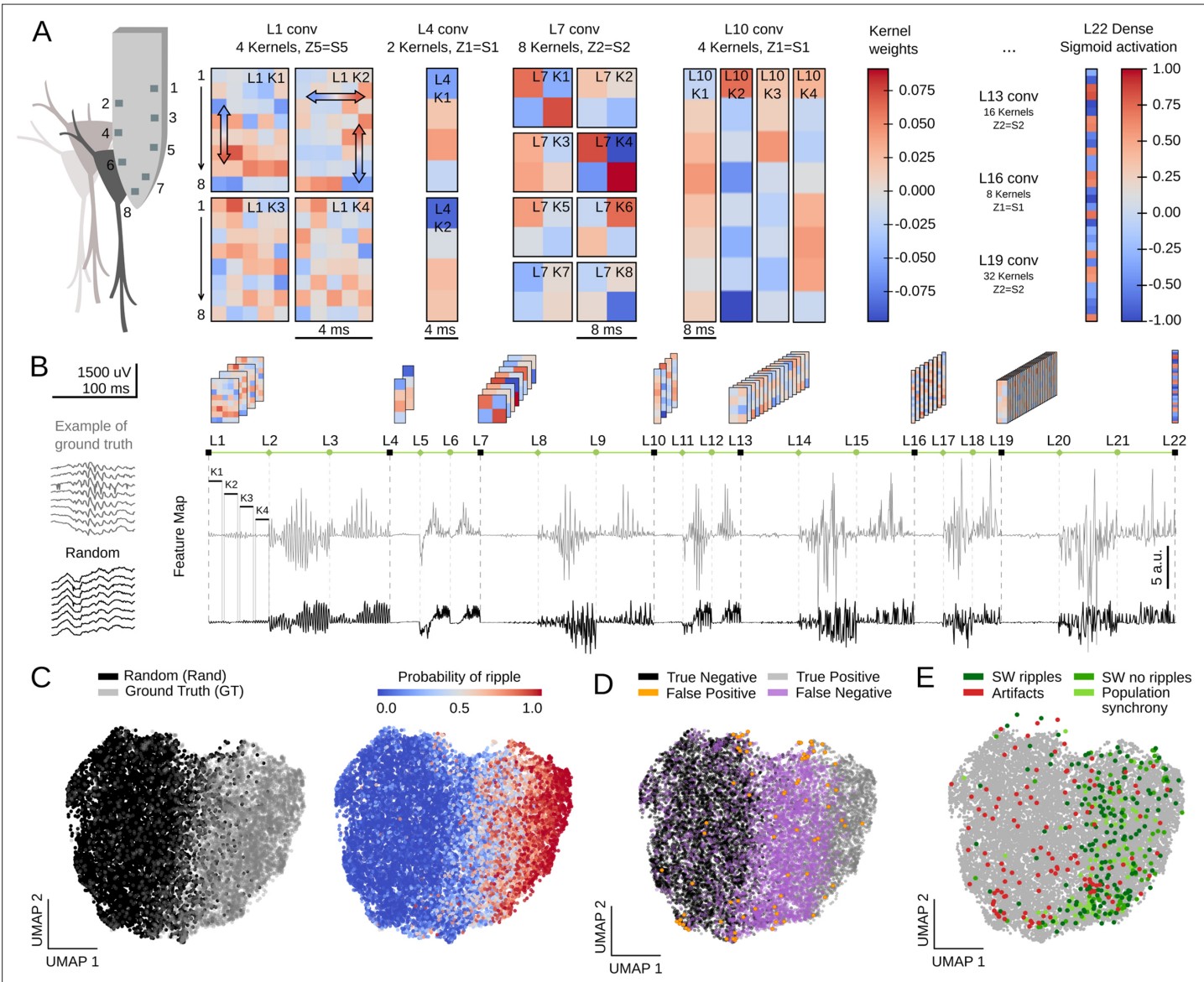

**Figure 6.** Feature map analysis of CNN32 operation. (**A**) Examples of kernel weights from different layers of CNN32. Note different distribution of positive and negative weights. In layer 1, the four different kernels act to transform the 8-channels input into a single channel output by differently weighting contribution across the spatial (upper and lower local field potential [LFP] channels; vertical arrows in L1K1 and L1K2) and temporal scales (horizontal arrow in L1K2). See the resulting kernel activation for the example sharp-wave ripple (SWR) event in *Figure 1D*. (**B**) Feature map from the example SWR event (100 ms window; gray) built by concatenating the kernel activation signals from all layers into a single vector. The feature map of a randomly selected LFP epoch without annotated SWR is shown at bottom (black). (**C**) Two-dimensional reduced visualization of CNN32 feature maps using Uniform Manifold Approximation and Projection (UMAP) shows clear segregation between similar number of SWRs (ground truth [GT]) and randomly chosen LFP epochs (Rand) (7491 events, sampled from 17 sessions, seven mice). Note distribution of SWR probability at right consistent with the ground truth. (**D**) Distribution of True Positive, True Negative, False Positive, and False Negative events in the UMAP cloud. (**E**) Distribution of the False Positive events previously validated in *Figure 4F*. Note that they all lay over the ground truth region.

The online version of this article includes the following figure supplement(s) for figure 6:

**Figure supplement 1.** Feature map analysis of CNN12 operation.

First, we examined the weights of the first layer kernels, which act directly over high-density LFP inputs. We noted that their profiles were especially suited for assessing critical LFP features, such as the laminar organization of activity. For example, L1K1 acted along the spatial scale by differentially weighting LFP channels along the somatodendritic axis and deep-superficial layers (*Figure 6A*), consistent with the saliency map shown above. In contrast, weights from L1K2 likely operated in the temporal scale with major differences along the kernel length (*Figure 6A*). In this case, by positively weighting upper channels at later samples this filter may be anticipating some SWR motifs, as shown before. Interestingly, opposing trends between top and bottom channels suggest some spatial effect as well. L1K3 and L1K4 provided less obvious integration across the spatial and temporal scales. In spite of the complexity of the resulting convolution along the entire event, visualization of KA reflects detection of ripples as well as the slow and fast deflections of the associated sharp wave (see L1 outputs in *Figure 1D* for CNN32; *Figure 6—figure supplement 1A,B* for CNN12).

The same reasoning applies to the next layers. However, since CNN acts to transform an LFP 'object' into a probability value, the spatial and temporal features of SWR events become increasingly abstract. Notwithstanding, their main features are still recognized. For example, L4K1 and L4K2 outputs likely reflected the spatiotemporal organization of the input SWR event, in particular the slower components and uneven distribution of ripples (see *Figure 1D* and *Figure 6—figure supplement 1A*).

To quantify these observations, we evaluated how the different kernels were activated by a similar number of LFP events centered at either the ground truth or at a random timing (*Figure 6B*, 7491 events in each category; data from both the training and test offline sessions). For each window, we concatenated the resulting KA from all layers in a single vector, dubbed feature map (*Figure 6B*; length 1329 for CNN32, 3991 for CNN12). Since each layer generates a characteristic activity in response to input data, we reasoned that feature maps should carry information on the network representation of a particular LFP event.

We used Uniform Manifold Approximation and Projection (UMAP), a computationally efficient dimensionality reduction and visualization tool, to explore feature maps. UMAP successfully segregated feature maps of LFP events according to their detection probability in a two-dimensional cloud (*Figure 6C*; *Figure 6—figure supplement 1C*), supporting that the entire CNN is coding for different features of SWR across layers (*Figure 6—figure supplement 1D*).

We labeled each LFP event in UMAP coordinates as True Positive (detected ground truth events), False Positive (random events detected as SWR), False Negative (undetected ground truth), and True Negative (unannotated and undetected events). We found striking segregation across the UMAP cloud with True Positive and True Negative events falling apart (*Figure 6D*; *Figure 6—figure supplement 1C*). False Negatives were mostly located at the intermediate region, suggesting they could be detected with less conservative thresholds. Interestingly, False Positive predictions were scattered all around the cloud, supporting the idea that they reflect heterogeneous events as seen above. Mapping all the previously validated False Positive events (see *Figure 4F*) over the UMAP cloud confirmed that those corresponding to population firing synchrony and sharp waves without ripples distributed over the ground truth, while those corresponding to artifacts mostly fell apart (*Figure 6E*).

Altogether, these analyses permitted us to understand how the CNN operates to detect SWR events. Our study suggests that a CNN relying on feature-based detection allows to capture a large diversity of SWR events. The new method, in combination with community tagging efforts and optimized filters, could potentially facilitate discovery and interpretation of the complex neurophysiological processes underlying SWR.

## Leveraging CNN capabilities to interpret SWR dynamics

Equipped with this tool we sought to understand the dynamics of SWR across the entire hippocampus. To this purpose, we obtained Neuropixels recordings from different rostro-caudal penetrations in head-fixed mice (*Figure 7A*; n=4 sessions, four mice; *Supplementary file 1*). Detailed post hoc histological analysis validated the probe tracks passing through a diversity of brain regions, including several thalamic nuclei as well as the dorsal and ventral hippocampus (*Figure 7B*, *Figure 7—figure supplement 1A*).

By exploiting the ultra-dense configuration of Neuropixels, we simulated consecutive penetrations covering the entire dorsoventral axis (*Figure 7A*). We run offline detection using eight neighboring

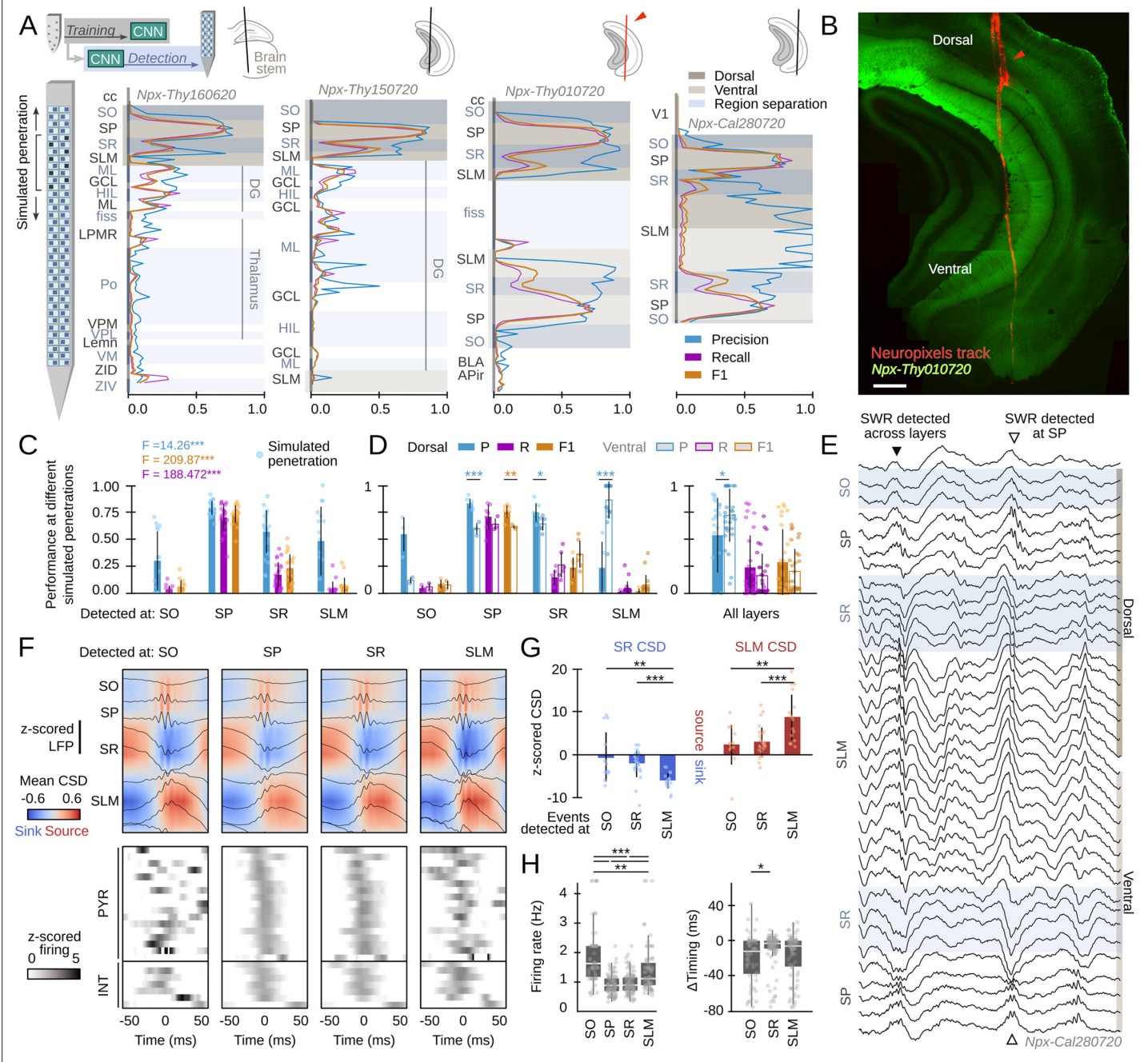

**Figure 7.** Hippocampus-wide sharp-wave ripple (SWR) dynamics through the lenses of convolutional neural network (CNN). (**A**) Neuropixels probes were used to obtain ultra-dense local field potential (LFP) recordings across the entire hippocampus. Offline detection was applied over continuous simulated penetrations (8-channels). Detection performance is evaluated across brain regions and hippocampal layers using the CNN trained with a different electrode type. See Methods for the list of acronyms. (**B**) Histological validation of one of the experiments shown in A (red arrowhead). Scale bar corresponds to 350 µm. (**C**) Performance of CNN32 across hippocampal layers (96 dorsal simulated penetrations, four mice). The results of an independent one-way ANOVA for P, R, and F1 is shown separately. ***, p<0.001. (**D**) Dorsoventral differences of CNN32 performance across layers. P, R, and F1 values from dorsal and ventral detection were compared pairwise (55 dorsal and 55 ventral simulated penetrations, four mice). *, p<0.05; **, p<0.01; ***, p<0.001. (**E**) Example of an SWR detected across several layers (black arrowhead). Note ripple oscillations all along the SR and SLM. A SWR event which was only detected at SP dorsal and ventral is shown at right (open arrowhead). (**F**) Mean LFP and current-source density (CSD) signals from the events detected at different layers of the dorsal hippocampus of mouse Npx-Thy160620 (top). Bottom plots show the SWR-triggered average responses of pyramidal cells and interneurons. Cells are sorted by their timing during SWR events detected at SP. (**G**) Quantification of the magnitude of the SR sink and SLM source for events detected at SO, SR, and SLM, as compared against SP detection. One-way ANOVA SR CSD: F(2)=9.13, p=0.0004;

*Figure 7 continued on next page*

*Figure 7 continued*

SLM CSD: F(2)=9.64, p=0.0003; **, p<0.01; ***, p<0.001. (**H**) Quantification of changes of firing rate and timing of pyramidal cells during SWR detected at different layers. Firing rate: F(3) = 28.68, p<0.0001; *, p<0.05; ***, p<0.001. Timing: F(2) = 10.18, p<0.0001; ***, p<0.0001.

The online version of this article includes the following figure supplement(s) for figure 7:

**Figure supplement 1.** Convolutional neural network (CNN) detection of sharp-wave ripple (SWR) from ultra-dense Neuropixels recordings.

Neuropixels channels as the inputs, then move four channels downward/upward and repeat detection again, up to the end of the probe. We used the original CNN32 without retraining, the Butterworth filter and RippleNET, to evaluate detection performance against the ground truth.

Consistent with data above, we found successful detection of SWR events by the CNN from the dorsal CA1 region (*Figure 7A*). While detection was optimal at the CA1 cell layer (stratum pyramidale [SP]), we noted many events were actually identified from SWR-associated LFP signatures at the radiatum (SR) and lacunosum moleculare (SLM) layers (*Figure 7C*; *Figure 7—figure supplement 1B*). When evaluated per layer, detection of SWR was better at the dorsal than at the ventral hippocampus, except for SR and SLM (*Figure 7D*, left). We found no major differences except for precision, when all layers were pooled together (*Figure 7D*, right). No difference in the rate of False Positives between SO (0.52±0.21), SR (0.50±0.21), and SLM (0.46±0.19) can account for this effect.

In spite that only a subset of SWR could be identified from recordings at SR and SLM (i.e. R-values were low), precision was very high (i.e. over 80% of predictions were consistent with the ground truth). A close examination of the morphology of these events confirmed they exhibited LFP and oscillatory features consistent with the kernel saliency maps (*Figure 7E*, *Figure 7—figure supplement 1C*). Remarkably, both the Butterworth filter and RippleNET failed to identify SWR-associated signatures beyond the dorsal SP (*Figure 7—figure supplement 1D,E*).

To gain insights into the underlying physiology and to discard for potential volume conduction effects, we simulated linear penetrations through the dorsal hippocampus and estimated the associated current-source density (CSD) signals of events detected at different layers (*Figure 7F*, top). We found larger sinks and sources for SWR that can be detected at SLM and SR versus those detected at SO (*Figure 7G*; z-scored by mean values of SWR detected at SP only). We also exploited Neuropixels to isolate activity from putative pyramidal cells (n=99) and interneurons (n=29, all penetrations) during the different SWR event types (*Figure 7F*, bottom). For pyramidal cells, we found striking reorganization of the firing rate and timing during SWR detected at SO, SR, and SLM (*Figure 7H*). Interneurons exhibited similar variability (*Figure 7—figure supplement 1E*). Timing and rate differences of pyramidal cell and interneuronal firing with respect to SWR events detected at different layers support the idea that they reflect activation of different hippocampal ensembles. Our CNNs thus provide unique opportunities to study the so far elusive dynamics accompanying SWR responses.

## Discussion

Here, we report how one-dimensional convolutional networks operating over high-density LFP recordings allows for improved detection and interpretation of hippocampal SWR events. While the network was trained in a subset of LFP data recorded around the dorsal CA1 cell layer of head-fixed mice, detection generalized across strata, brain locations (e.g. ventral hippocampus), preparations (i.e. freely moving), and species (i.e. rats) without the need for retraining. Our CNN exhibited a much higher stability, less threshold-dependent sensitivity, and overall higher performance as compared with the spectral filter and RippleNET, a recurrent neural network solution. This unique capability of our CNN relies on feature-based analysis of LFP signals, which provide similar explanatory power as standard LFP profiling. Such a developmental potential of convolutional neural networks permits challenging the interpretation of brain signals (*Frey et al., 2021*), and SWR in particular (this study).

From a physiological perspective, studying brain function relies in understanding activity in relation to behavior and cognition (*Cohen, 2017*; *Friston et al., 2015*). Inspired by the tradition to observe and categorize, neuroscientists require classifying EEG/LFP signals into patterns, which presumably should gain mechanistic significance at the neuronal and microcircuit levels (*Buzsáki and Draguhn, 2004*; *Fernández et al., 1999*; *Niedermeyer and Silva, 2005*). Yet, some of the most widely used classification schemes still generate debate. For instance, contributors to gamma oscillations (40–100 Hz) include fluctuating synaptic potentials reflecting inhibition, excitation, or

both in interaction with phase-locking firing from subsets of cells (*Atallah and Scanziani, 2009*; *Bartos et al., 2007*). The specific contribution of the different factors at the resulting dominant oscillatory frequency band is not trivial (*Buzsáki and Schomburg, 2015*). In addition, relying on spectral definitions to analyze EEG/LFP data has to cope with the nonstationary nature of brain activity, while the demarcation of frequency bands does not necessarily fit to unambiguous basic mechanisms. Whether this reflects the elusive emergent behavior of brain activity or methodological limitations is arguable.

We aimed exploiting machine-learning tools to transform the study of hippocampal SWR, a major neurophysiological event underlying memory trace consolidation and recall (*Buzsáki, 2015*). While SWR presumably entail coordinated activity of pyramidal cells and GABAergic interneurons in a mnemonically relevant sequence-specific manner (*Diba and Buzsáki, 2007*; *Gridchyn et al., 2020*; *Olafsdóttir et al., 2018*; *Stark et al., 2015*; *van de Ven et al., 2016*), their physiological definition seems constrained (*Buzsáki and Schomburg, 2015*). Moreover, the replay content and order unfold neuronal representations in a myriad of combinations in the service for cognitive agency and flexibility (*Joo and Frank, 2018*; *Pfeiffer, 2020*). The potentially different mechanisms underlying such a representational complexity are not yet integrated into the existing definition of SWR (*de la Prida, 2020*).

When coupled to ultra-dense Neuropixels, our CNN identified subsets of SWR across different strata of the dorsal and ventral hippocampus. The ability to detect events across layers seemed to rely in a combination of features with the strength and visibility of the associated current sinks/sources having major contributions. This calls for the existence of different generators emerging from interaction between different input pathways and local microcircuits (*de la Prida et al., 2006*). For instance, recent data suggest pivotal role of entorhinal inputs in modulating and elongating the dynamics of locally generated SWRs (*Fernández-Ruiz et al., 2019*; *Oliva et al., 2018*; *Yamamoto and Tonegawa, 2017*). Similarly, SWR events disproportionally weighted by downstream inputs along the CA3 to CA2 axis differentially modulate consolidation of recognition memory at the social and nonsocial domains (*Nakashiba et al., 2009*; *Oliva et al., 2020*; *Oliva et al., 2016*). Consistently, we found that some ripples can be actually detected at SO, SR, and SLM strata independently on their alleged local generation at the CA1 cell layer.

The configuration of the current sinks and sources associated to independently detected SWR events suggest that the weighted interaction between fluctuating input pathways may entail contribution by different factors across behavioral states (*Buzsáki, 2015*). For instance, different subcircuits may contribute to sleep and awake SWR with different cognitive roles (*Roumis and Frank, 2015*). The ability to detect ripple oscillations at different layers also indicate a role for dendritic potentials, such as complex spikes and dendritic bursts (*Bittner et al., 2015*; *Kamondi et al., 1998*). Finally, while attention is traditionally focused on parvalbumin and cholecystokinin GABAergic basket cells providing perisomatic innervation (*Klausberger et al., 2005*), other GABAergic cells and terminals located at the border between SR and SLM may equally contribute (*Basu et al., 2016*; *Kitamura et al., 2015*; *Klausberger and Somogyi, 2008*). This is supported by larger current sources associated with SWR events detected at SLM layers, as we show here.

Our data suggest that only one part of the dorsal SWR dynamics can be explained locally, consistent with complex interaction along the septotemporal axis (*Patel et al., 2013*). Instead, the CNN identify different types of SWR events detected at distant strata suggesting major role of input pathways. A segregated role for dorsal and ventral SWR events suggest that brain-wide subcircuits inherit the different representational dynamics of a variety of replays (*Sosa et al., 2020*). The detection unfolding of CNN thus permit an unbiased categorization without relying on more elusive spectral criteria. Critically, both the filter and RippleNET failed to capture SWR diversity across strata further confirming the suitability of CNN to identify critical LFP features accompanying a wealth of events.

Our method also identified events beyond the individual expert ground truth. Careful examination of those False Positives reveal sharp waves associated to population firing without ripples, as well as other unclassified forms of activities. While we cannot discard noisy detection from a continuum of LFP activity, our categorization suggest that they may reflect processes underlying buildup of population events (*de la Prida et al., 2006*). In addition, the ability of CA3 inputs to bring about gamma oscillations and multi-unit firing associated with sharp waves is already recognized (*Sullivan et al., 2011*), and variability of the ripple power can be related with different cortical subnetworks (*Karimi Abadchi et al., 2020*; *Ramirez-Villegas et al., 2015*). Since the power spectral level operationally defines the

detection of SWR, part of this microcircuit intrinsic variability may be escaping analysis when using spectral filters.

Understanding how the brain encodes for memory is challenging. Recent data suggest that replay emerging from SWR is more complex than originally thought (*Joo and Frank, 2018*). Cell-type-specific subcircuits, operating over a variety of interneuronal classes and under the influence of different input pathways, provide mechanistic support for a wealth of SWR events (*de la Prida, 2020*). Yet, SWR detected by gold-standard spectral methods fail to reflect the necessary statistical variance that allows for identifying specific trends. Relying on unbiased feature-based methods hopefully can change the game.

## Materials and methods
### Animals
All protocols and procedures were performed according to the Spanish legislation (RD 1201/2005 and L.32/2007) and the European Communities Council Directive 2003 (2003/65/CE). Experiments were approved by the Ethics Committee of the Instituto Cajal and the Spanish Research Council.

In this work, we used different mouse lines aimed to target different cell-type-specific populations for optogenetic and imaging experiments. Experiments included in this paper follow the principle of reduction, to minimize the number of animals used and this is the reason why we obtained data from different mouse lines. Animals and sessions used are summarized in *Supplementary file 1*. Animals were maintained in a 12 hr light-dark cycle (7 a.m. to 7 p.m.) with access to food and drink ad libitum.

### Head-fixed preparation
Mice were implanted with fixation bars under isoflurane (1.5–2%; 30% oxygen) anesthesia. Bars and ground/reference screws (over the cerebellum) were fixed with light-curing acrylic resins (OptiBond and UNIFAST LC). After surgery, mice were treated with buprenorphine during 2 days. For opto-genetic experiments, mice from different promotor-specific Cre lines were previously injected with AAV5-DIO-EF1a-hChR2-EYFP (1 µl; titer 4.5×1012 vg/ml; provided by UNC Vector core, Deisseroth lab) targeting the dorsal CA1 region (−1.9 mm AP; 1.25 mm ML and 1 mm depth). Transgenic Thy1-ChR2-YFP and Thy1-GCaMP7 mice were directly implanted with fixation bars.

Two days after surgery, mice were habituated to head-fixed conditions (10–14 days of training). The apparatus consisted on a wheel (40 cm diameter) hosting different somatosensory cues and equipped with a Hall sensor (HAMLIN 55300; Littelfuse Inc) to track for position analogically. Animals were water rewarded just after each training session (2–4 sessions ×day). After several days, mice were able to stay comfortable in the apparatus with periods of running, grooming, and immobility.

Once habituated, mice were anesthetized with isoflurane and a craniotomy was practiced for electrophysiological recordings (antero-posterior: −3.9 to −6 mm from Bregma; medio-lateral: 2–5 mm). The craniotomy was sealed with Kwik-Cast silicone elastomer and mice returned to their home cage. Recording sessions started the day after craniotomy.

### Electrophysiological recordings
LFP recordings were obtained with integrated µLED optoelectrodes (32 channels, 4 shanks of 8-channels, and 3 µLED each) originally provided by Euisik Yoon under the NSF-funded NeuroNex project and later purchased from NeuroLight Technologies, LLC, N1-A0-036/18 and Plexon. Wide-band (1 Hz–5 KHz) LFP signals were recorded at 30 KHz sampling rate with an RHD2000 Intan USB Board running under OE. Optoelectrode recordings targeted the dorsal CA1 region, using characteristic features such as the laminar profile of theta and SWRs, as well as unit activity to infer position within the hippocampus.

Ultra-dense recordings were obtained with Neuropixels 1.0 probes and acquired with the PXIe acquisition module mounted in the PXI-Express chassis (National Instruments). Neuropixels probes consist of up to 966 recording sites (70×20 µm) organized in a checkerboard pattern, from which 384 can be selected for recording. Recordings were made in external reference mode with LFP gain set at 250 and at 2500 Hz sampling rate, using the SpikeGLX software. The probe targeted the dorsal-to-ventral hippocampus at different anterior-to-posterior positions in four different mice (*Supplementary*

*file 1*). To recover the penetrating track precisely, the back of the Neuropixels probe was coated with DiI (Invitrogen).

After completing experiments, mice were perfused with 4% paraformaldehyde and 15% saturated picric acid in 0.1 M (pH 7.4) phosphate-buffered saline (PBS). Brains were post-fixed overnight, washed in PBS, and serially cut in 70 μm coronal sections (Leica VT 1000S vibratome). Sections containing the probe tracks were identified with a stereomicroscope (S8APO, Leica) and mounted on glass slides in Mowiol (17% polyvinyl alcohol 4–88, 33% glycerin, and 2% thimerosal in PBS).

Sections from Neuropixels recording were analyzed with SHARP-Track, a tool to localize regions going through electrode tracks (*Shamash et al., 2018*) (https://github.com/petersaj/AP_histology *Peters, 2022*). Acronyms in *Figure 7* correspond to the following: corpus callosum (cc); primary visual cortex (V1); stratum oriens (SO); stratum piramidale (SP); stratum radiatum (SR); stratum lacunosum moleculare (SLM); molecular layer dentate gyrus (ML); granular layer dentate gyrus (GCL); hilus (HIL); hippocampal fissure (fiss); basolateral amygdala (BLA); amygdalopiriform transition area (APir); lateral posterior medial rostral thalamus (LPMR); posterior thalamus (Po); ventro posterior medial thalamus (VPM); ventro posterior lateral thalamus (VPL); lemniscus (Lemn); ventro medial thalamus (VM); zona incerta, dorsal part (ZID); zona incerta, ventral part (ZIV).

## Neural network specifications

We used Python 3.7.9 with libraries Numpy 1.18.5, Scipy 1.5.4, Pandas 1.1.4, and H5Py 2.10.0 for programming different routines. To build, train, and test the network, we use the Tensorflow 2.3.1 library, with its built-in Keras 2.4.0 application programming interface (API). Training and offline validation of the CNN was performed over the Artemisa high-performance computing infrastructure (https://artemisa.ific.uv.es/web/content/nvidia-tesla-volta-v100-sxm2). It consisted in 23 machines equipped with four NVIDIA Volta V100 GPUs. Analyses were conducted on personal computers (Intel Xeon E3 v5 processor with 64 GB RAM and Ubuntu v.20.04).

The CNN architecture was designed as a sequence of blocks integrated by one one-dimensional Convolutional layer (*Cun et al., 1990*) followed by one BatchNorm layer (*Ioffe and Szegedy, 2015*), and one Leaky ReLU Activation layer (*Maas et al., 2013*). There were seven of these blocks (21 layers) and a final Dense layer (*Rosenblatt, 1958*) (layer 22) (*Figure 1C*).

One-dimensional Convolutional layers (tf.keras.layers.Conv1D) were in charge of processing data and looking for characteristic features. These layers have a determined number of kernels, which was defined in the parameter search. A kernel is a matrix of weights acting to apply a convolution operation over data. The result of this operation is known as the KA signal. A Convolutional layer generates as many KA as the number of kernels it has. BatchNorm layers (tf.keras.layers.BatchNormalization) perform a normalization of the Convolutional layer KA, fixing its means and variances and providing stability and robustness to the whole network. Leaky ReLU layer (tf.keras.layers.LeakyReLU) has a similar purpose to the BatchNorm layer, making the network more stable by transforming negative input values into numbers very close to 0. The final Dense layer (tf.keras.layers.Dense) was fit to the dimension of the output space (i.e. probability values).

BatchNorm layer parameters were all left as their default values defined in the Tensorflow 2.3.1 library. The Leaky ReLU layer alpha parameter was set to 0.1. For Convolutional layers, the kernel size and stride were set to the same value so that the network operates similarly offline and online. The kernel size and stride determined the duration of the input window, so they were tuned in order to fit either a 32 ms window (CNN32) or a 12.8 ms window (CNN12). Values of kernel size and stride for Convolutional layers 1, 4, 7, 10, 13, 16, and 19 of CNN32 were: 5, 1, 2, 1, 2, 1, 2, respectively. For CNN12, the values were 2, 1, 2, 1, 2, 1, and 2. Since max-pooling layers can be replaced by Convolutional layers with increased stride, we chose not using max-pooling layers to avoid issues with the input window size (*Springenberg et al., 2014*).

The number of kernels and kernel regularizers were selected after performing an initial parametric search (initial learning rate, number of kernels factor, and batch size; *Figure 1—figure supplement 1E*). For the Dense layer we used a sigmoid activation function operating over 1 unit to produce 1 channel output. All the other parameters for the Convolutional layers, as well as for the Dense layer, were set initially at default values. Additionally, we also tested whether adding two LSTM layers before the final Dense layer improved performance in the preliminary parameter tests.

We selected our CNN32 as that with the lower and more stable training evolution (see below) from the 10-best networks in the initial parameter search (out of 107; *Supplementary file 1B*). CNN32+LSTM networks exhibited similar performance, but took substantially more time for training. A more thorough parametric search was conducted over a larger set of parameters (initial learning rate, number of kernels factor, batch size, optimizer, optimizer epsilon, regularizer, regularizer value, and decay; *Figure 1—figure supplement 1E*) for both types of architectures. The initially selected CNN32 was among the 30-best networks of the extended parameter search, also exhibiting fast training and high loss evolution stability (out of 781). Based on parametric searches, we chose the Adam algorithm as the optimizer (*Kingma and Ba, 2015*) with an initial learning rate of 0.001, beta_1=0.9, beta_2=0.999, and epsilon = 1e-07. Batch size was set at 16 and the number of kernels for Convolutional layers 1, 4, 7, 10, 13, 16, and 19 was set at 4, 2, 8, 4, 16, 8, and 32, respectively. A L2 regularization method (*Tikhonov and Arsenin, 1977*) with a 0.0005 value was employed to avoid overfitting. No additional learning rate decay was used.

## Ground truth and data annotation

A MATLAB R2019b tool was designed to annotate and validate data by an expert electrophysiologist (original expert). All data was visually inspected and SWR events annotated. An important decision we made was to manually annotate the start and the end of SWR events so that the network could learn anticipating events in advance. The start of the event was defined near the first ripple or the sharp-wave onset. The end of the event was defined at the latest ripple or when sharp-wave resumed. While there was some level of ambiguity on these definitions, we opted for including these marks in order to facilitate transition to ground truth detection. An additional expert (new expert) tagged SWR independently using a subset of sessions from the offline validation and online pool, to allow for comparisons between experts in the same lab.

## Data preparation

Datasets used for training and development of the CNN were created by loading a number of experimental sessions and storing them in two different three-dimensional matrices, X and Y.

Matrix X stored several chunks of 8-channels LFP recordings. From each session, LFP data from all probe shanks displaying any SWR were loaded, unless specific shanks were selected. If a shank had more than 8 channels, then they were randomly selected while giving priority to those located at the SP of CA1. All LFP signals were down-sampled to 1250 Hz and normalized using z-score. LFP signals were sliced into chunks of 57.6 s, which is exactly divisible by 0.032 and 0.0128 s, in order to keep a consistent matrix shape even when various sessions of different durations were used. This chunk size maintains the properties of long duration signals, which is essential for the CNN to reach a high-performance score when fed with continuous data. Chunks with no SWR events would be discarded, but that was an extremely rare case. At the end of this process the result is a matrix X with dimension (n, 72,000, 8), where n is the number of chunks of 72,000 samples (57.6 s sampled at 1250 Hz) for each of the eight LFP channels.

Matrix Y contained the annotated labels for each temporal window (32 or 12.8 ms) stored in X. To create Y, each chunk was separated in windows of 32 or 12.8 ms and then assigned a label, a number between 0 and 1, depending on the percentage of the window occupied by a SWR event. Therefore, dimension of matrix Y was (n, 1800, 1) for CNN32, since there are 1800 32 ms windows in a chunk of 57.6 s, and (n, 4500, 1) for CNN12, with 4500 windows of size 12.8 ms for each chunk.

Finally, the whole dataset (both X and Y matrices) was separated into the training set, used to fit the model, and the development set, used to evaluate the performance of the trained model with different data than those used for training while still tuning the network hyper-parameters. Train set took 70% of the data and development set the remaining 30%.

## CNN training, development, and testing

Two sessions from two different mice were used as the training set (*Supplementary file 1*). Training was run for 3000 epochs using the binary cross-entropy as loss function:

$$H_P(q) = -\frac{1}{N}\sum_{i=1}^{N} y_i \cdot \log\left(p\left(y_i\right)\right) + \left(1 - y_i\right) \cdot \log\left(1 - p\left(y_i\right)\right)$$

where N is the number of windows, $y_i$ is the label of window I, and $p(y_i)$ is the probability predicted for window i.

In order to evaluate the network performance, two different datasets were used: the training set described above, and the validation set, consisting of 15 sessions from five different animals that were not used for training or development (***Supplementary file 1***).

To detect SWR event, we set a probability threshold to identify windows with positive and negative predictions. Accordingly, predictions were classified in four categories: True Positive (TP) when the prediction was positive and the ground truth window did contain an SWR event; False Positive (FP) when the prediction was positive in a window that did not contain any SWR; False Negative (FN) when the prediction is negative but the window contained a SWR; and True Negative (TN) when the prediction was negative and the window did not contain any SWR event.

Intersection over Union (IOU) methodology was employed to classify predictions into those four categories. It was calculated by dividing the intersection (overlapping) of two windows by the union of them:

$$IoU = \frac{window_1 \cap window_2}{window_1 \cup window_2}$$

Two windows were considered to match if their IOU was equal to or greater than 0.1. If a positive prediction had a match with any window containing a ripple it was considered a TP, or it was classified as FP otherwise. All true events that did not have any matching positive prediction were considered FN. Negative predictions with no matching true events windows were TN.

With predicted and true events classified into those four categories there are three measures than can be used to evaluate the performance of the model. Precision (P), which was computed as the total number of TPs divided by TPs and FPs, represents the percentage of predictions that were correct.

$$Precision = \frac{TP}{TP+FP}$$

Recall (R), which was calculated as TPs divided by TPs and FNs, represents that percentage of true events that were correctly predicted.

$$Recall = \frac{TP}{TP+FN}$$

Finally, the F1 score, calculated as the harmonic mean of Precision and Recall, represents the network performance.

$$F1 = \frac{2*(Precision*Recall)}{Precision+Recall}$$

As mentioned before, a prediction was considered positive when its probability surpassed a specified threshold. During offline detection, a first threshold was used to indicate the potential onset of the SWR event, followed by a second confirmatory higher threshold, which identifies the event itself. In order to select the best thresholds for offline validation, all combinations were compared and the one that yielded the best F1 score was chosen. Possible values for the first threshold were 0.80, 0.75, 0.70, 0.65, 0.60, 0.55, 0.50, 0.45, 0.40, 0.35, 0.30, 0.25, 0.20, 0.15, and 0.10, while for the second threshold were 0.80, 0.70, 0.60, 0.50, 0.40, 0.30, 0.20, and 0.10. Note that only the higher threshold is the one that defines detection, which is reported in figures. For online detection, only one threshold was used and it was adjusted manually at the beginning of the experiment based on the expert criteria.

To estimate delay between prediction and SWR events, the temporal relation between correct predictions and their matching true events was measured. SWR ripple peaks were defined after filtering the relevant LFP channel using a third-order Butterworth filter and an enlarged bandpass between 70 and 400 Hz. The resulting signal was subsequently filtered with a fourth-order Savitzky-Golay filter and smoothed twice with windows of 3 and 6.5 ms to obtain the SWR envelope. The maximal value of the envelope signal was defined as the SWR ripple peak. The interval between the initial prediction time and the SWR ripple peak was defined as the time to peak.

The trained model is accessible at the Github repository for both Python: https://github.com/PridaLab/cnn-ripple, (copy archived at swh:1:rev:9dcc5b6a8267b89eb86a2813dbbcb74a621a701b; ***Amaducci and Navas-Olive, 2021***) and MATLAB: https://github.com/PridaLab/cnn-matlab (copy

archived at swh:1:rev:060b2ff6e4b6c5eacb9799addd5123ad06eaaf33; *Navas-Olive and Esparza, 2022*). Code visualization and detection is shown in an interactive notebook https://colab.research.google.com/github/PridaLab/cnn-ripple/blob/main/src/notebooks/cnn-example.ipynb.

## Offline detection of SWR events with Butterworth filters

Standard ripple detection tools are based on spectral filters. In order to compare online and offline performance, we adopted the OE bandpass (100–300 Hz passband) second-order Butterworth filter as the gold standard. We confirmed that the choosen filter parameters provided optimal performance when tested with the same training set used for the CNN. Offline filter detection was computed in MATLAB R2019b, using the *Butterworth* filter (100–300 Hz passband) and a non-casual filter *filtfilt* to avoid phase lags. In order to compute the envelope, the filtered signal was amplified twice, filtered by a fourth-order Savitzky-Golay filter, and then smoothed by two consecutive *movmean* sliding windows (2.3 and 6.7 ms). A detection had to fulfill two conditions: the envelope had to surpass a first threshold, which will define the ripple beginning and end, and a second threshold to be considered a detection. Detections closer than 15 ms were merged. Performance of ripple detection methods is very sensitive to the chosen threshold. To look for the fairest comparison, we made predictions for all possible combinations of the first threshold being 1, 1.5, 2, 2.5 times the envelope standard deviation, and the second threshold being 3, 3.5, 4, 4.5, 5, 5.5, 6, 6.5, 7, 7.5, 8, 8.5, 9, 9.5, 10 times the envelope standard deviation (giving a total of 60 threshold combinations). We then chose the one that scored the maximum F1. This was done separately for each session.

Online detections were defined whenever the filtered signal was above a single threshold (see next section). To exclude for artifacts and to cope with detection standards (*Fernández-Ruiz et al., 2019*), an additional non-ripple channel was used to veto high-frequency noise detections.

## Characterization of SWRs

Ripple properties were computed using a 100 ms window around the center of the event of the pyramidal channel of the raw LFP. Preferred frequency was computed first by calculating the power spectrum of the 100 ms interval using the enlarged bandpass filter 70 and 400 Hz, and then looking for the frequency of the maximum power. In order to account for the exponential power decay in higher frequencies, we subtracted a fitted exponential curve ('fitnlm' from MATLAB toolbox) before looking for the preferred frequency. The <100 Hz contribution was computed as the sum of the power values for all frequencies lower than 100 Hz normalized by the sum of all power values for all frequencies (of note, no subtraction was applied to this power spectrum). The entropy was computed using a normalized power spectrum (divided by the sum of all power values along all frequencies) as:

$$Entropy = -\sum Power\left(f\right) \cdot log_2\left(Power\left(f\right)\right)$$

## OE custom plugins for online detection

Two custom plugins were developed using OE GUI 0.4.6 software in a personal computer. The first plugin was designed to detect when a signal crossed a determined amplitude threshold, defined as the signal standard deviation multiplied by some number. It was used in combination with the Band-pass Filter plugin, which implements a Butterworth filter, so the input for the crossing detector was a filtered signal. In order to avoid artifacts, we use a second input channel from a separate region defined by the experimenter. Events detected in both channels were discarded.

The second plugin was developed to operate the CNN and it used the Tensorflow 2.3.0 API for C (https://www.tensorflow.org/install/lang_c). Since the network was trained to work with data sampled at 1250 Hz, the plugin down-sampled the input channels. It also separated data into windows of 12.8 ms and 8-channels to feed into the CNN every 6.4 ms. Detection threshold was defined as a probability between 0 and 1, and it was manually adjusted by the experimenter.

Both plugins normalized the input data using z-score normalization. They require a short calibration time (about 1 min) to calculate the mean and standard deviation of the signals. The user could establish the detection thresholds and when an event is found a signal is sent through a selected output channel. For OE simulated experiments we used the same setup but the data was read from a file instead of streamed directly from the experiment. Events in simulated experiments were detected similarly as real-time experiments. Detection plugin: https://github.com/PridaLab/CNNRippleDetecto

rOEPlugin (copy archived at swh:1:rev:52b182d1fba732a0bc3ad69ce9453c6fe96ae190; *Esparza, 2022*).

## Closed-loop optogenetic experiments

For closed-loop experiments, the output channel from the OE plugin was fed into an Arduino board (Nano ATmega328) using an USB 3.0 connection. Optogenetic stimulation was performed with integrated µLED optoelectrodes using the OSC1Lite driver from NeuroLight Technologies controlled by the Arduino. Microwatt blue light stimulation at 10–20 µW was used to activate cell-type-specific ChR2. Specificity of viral expression and localization of probe tracks were histologically assessed after experiments.

## Computing the kernel saliency maps

During training, kernels weights are updated so each of them specializes in detecting a particular feature of SWR input data. In order to interpret these features, we adapted a methodology used in two-dimensional CNNs for image processing (*Simonyan et al., 2013*). First, we created an input 8×40 LFP signal with random values (standard normal distribution, with 0 mean and 0.01 standard deviation). Then, for each kernel we updated this input signal applying a stochastic gradient optimizer (tf.keras.optimizers.SGD) with a learning rate of 0.1, momentum 0.1, and a loss function equal to minus the normalized KA that produced such input (therefore making it gradient ascent). We repeated this optimization process until the mean squared error between the previous input and the optimized input was less than $10^{-9}$, or after 2000 iterations, whatever came first. The resulting input signal would be one that the chosen kernel is maximally responsive to. Code example for this process applied to the ResNet50V2 model of the ImageNet dataset (https://www.image-net.org/) can be found at the Keras documentation: https://keras.io/examples/vision/visualizing_what_convnets_learn/.

## Uniform Manifold Approximation and Projection

UMAP is a dimensional reduction technique commonly used for visualization of multi-dimensional data in a two- or three-dimensional embedding (*McInnes et al., 2018*). The embedding is found by searching a low dimensional projection with the closest equivalent fuzzy topological structure to that of the hyper-dimensional input data. We run UMAP version 0.5.1 (https://umap-learn.readthedocs.io/en/latest/) in Python 3.7 Anaconda.

We applied UMAP to decode CNN operation using the network feature maps in response to a diversity of LFP inputs. Feature maps were built by concatenating the resulting KA from all the Convolutional layers resulting in a 1329-dimensional vector (CNN32; 3991 for CNN12). The goal was to compute the reduced two-dimensional UMAP embedding from a large number of LFP events.

We computed the UMAP embedding of the feature map of the CNN using 7491 SWR events and 7491 random events and projected them in a color scale reflecting the different labels. Two-dimensional UMAP embeddings were evaluated for different parameter combinations of the number of neighbors and the minimal distance. After noticing no major differences, we choose their default values.

## Pattern matching

Pattern matching between saliency maps from the different kernels and the LFP windows was computed using *matchTemplate* from CV2 package (version 4.5.1), OpenCV library for python, with the *TM_CCORR* template matching operation. It slides a template (saliency map) along the whole signal (LFP window) and outputs a measure on their similarity for each slide. LFP windows provided were 100 ms 8-channel (8×125) z-scored windows around all true positive events, same number of true negative events, and all true positive and false positive events for both the training and validations sets. Windows were centered on the LFP minimum of the pyramidal channel closest to the maximum of the SWR envelope within a 10 ms window (envelope computed as described above).

## Simulated penetrations along Neuropixels probe

Simulated penetrations were obtained choosing Neuropixel electrodes with a relative distance similar to the µLED optoelectrode probe. To this purpose, we chose Neuropixel external electrodes (64 µm horizontal separation versus 70 µm for the µLED probe), alternating left and right for each row, so

the vertical distance was 20 μm (same as in μLED probes). Therefore, a simulated penetration always consisted of eight neighboring electrodes (e.g. [1 4 5 8 9 12 13 16]). To evaluate changes across layers and regions, the simulated penetration was moved all along the Neuropixels probe in 93 steps (downward/upward) thus providing a continuous mapping of LFP signals. For example, the following penetration sequence spanned along the brain: [1 4 5 8 9 12 13 16], [5 8 9 12 13 16 17 20], [9 12 13 16 17 20 21 24], and so on.

For CSD analysis we proceeded similarly, but choosing LFP channels every 100 μm to mimic a 16-channel silicon probe. CSD signals were calculated from the second spatial derivative. Smoothing was applied to CSD signals for visualization purposes only. Tissue conductivity was considered isotropic across layers.

### Quantification and statistical analysis

Statistical analysis was performed with Python 3.8.5 and/or MATLAB R2019b. No statistical method was used to predetermine sample sizes, which were similar to those reported elsewhere. Normality and homoscedasticity were confirmed with the Kolmogorov-Smirnov and Levene's tests, respectively. The number of replications is detailed in the text and figures.

Several ways ANOVAs were applied for group analysis. Post hoc comparisons were evaluated with the Tukey-Kramer test and whenever required Bonferroni correction was applied. For paired comparisons the Student's t-test was used. Correlation between variables was evaluated with the Pearson product-moment correlation coefficient, which was tested against 0 (i.e. no correlation was the null hypothesis) at $p < 0.05$ (two sided). In most cases values were z-scored (value – mean divided by the SD) to make data comparable between animals or across layers.

## Acknowledgements

This work is supported by grants from Fundación La Caixa (LCF/PR/HR21/52410030; DeepCode). Access to the Artemisa high-performance computing infrastructure (NeuroConvo project) is supported by Universidad de Valencia and co-funded by the European Union through the 2014–2020 FEDER Operative Programme (IDIFEDER/2018/048). ANO and RA are supported by PhD fellowships from the Spanish Ministry of Education (FPU17/03268) and Universidad Autónoma de Madrid (FPI-UAM-2017), respectively. We thank Elena Cid for help with histological confirmation of the probe tracks and Pablo Varona for feedback and discussion. We also thank Aarón Cuevas for clarifications and support while developing the Open Ephys Plugin for online detection.

## Additional information

### Competing interests

Liset M de la Prida: Reviewing editor, *eLife*. The other authors declare that no competing interests exist.

### Funding

| Funder | Grant reference number | Author |
| --- | --- | --- |
| Fundacion La Caixa | LCF/PR/HR21/52410030 | Liset M de la Prida |
| Ministerio de Educacion | FPU17/03268 | Andrea Navas-Olive |
| Universidad Autónoma de Madrid | FPI-UAM-2017 | Rodrigo Amaducci |

The funders had no role in study design, data collection and interpretation, or the decision to submit the work for publication.

### Author contributions

Andrea Navas-Olive, Rodrigo Amaducci, Conceptualization, Software, Formal analysis, Investigation, Visualization, Methodology, Writing – review and editing; Maria-Teresa Jurado-Parras, Data curation, Validation, Investigation, Writing – review and editing; Enrique R Sebastian, Software, Formal analysis;

Liset M de la Prida, Conceptualization, Supervision, Funding acquisition, Validation, Investigation, Writing - original draft, Project administration, Writing – review and editing

**Author ORCIDs**
Liset M de la Prida (ID) http://orcid.org/0000-0002-0160-6472

**Ethics**
All protocols and procedures were performed according to the Spanish legislation (R.D. 1201/2005 and L.32/2007) and the European Communities Council Directive 2003 (2003/65/CE). Experiments and procedures were approved by the Ethics Committee of the Instituto Cajal and the Spanish Research Council (PROEX131-16 and PROEX161-19). All surgical procedures were performed under isoflurane anesthesia and every effort was made to minimize suffering.

**Decision letter and Author response**
Decision letter https://doi.org/10.7554/eLife.77772.sa1
Author response https://doi.org/10.7554/eLife.77772.sa2

---

## Additional files

**Supplementary files**
• Supplementary file 1. Sessions and animals used for the different analysis.
• Transparent reporting form

**Data availability**
Data is deposited in the Figshare repository https://figshare.com/projects/cnn-ripple-data/117897. The trained model is accessible at the Github repository for both Python: https://github.com/PridaLab/cnn-ripple (copy archived at swh:1:rev:9dcc5b6a8267b89eb86a2813dbbcb74a621a701b) and Matlab: https://github.com/PridaLab/cnn-matlab (copy archived at swh:1:rev:060b2ff6e-4b6c5eacb9799addd5123ad06eaaf33). Code visualization and detection is shown in an interactive notebook https://colab.research.google.com/github/PridaLab/cnn-ripple/blob/main/src/notebooks/cnn-example.ipynb. The online detection Open Ephys plugin is accessible at the Github repository: https://github.com/PridaLab/CNNRippleDetectorOEPlugin (copy archived at swh:1:rev:52b182d1fba732a0bc3ad69ce9453c6fe96ae190).

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
