## [Editor Report]

This paper will be of interest to the neuroscience community studying brain oscillations. It presents a new method to detect sharp wave-ripples in the hippocampus with deep learning techniques, instead of the more traditional signal processing approach. The overall detection performance improves and this technique may help in identifying and characterizing previously undetected physiological events.

---

## [Decision Letter]

**Decision letter after peer review:**

Thank you for submitting your article "Deep learning based feature extraction for prediction and interpretation of sharp-wave ripples" for consideration by *eLife*. Your article has been reviewed by 2 peer reviewers, and the evaluation has been overseen by a Reviewing Editor and John Huguenard as the Senior Editor. The reviewers have opted to remain anonymous.

In the present study, Navas-Olive et al., introduce a novel method to detect and characterize sharp-wave ripples (SWRs) in the hippocampus. Specifically, the study presents how a convolutional neural net (CNN) may achieve better performance than more traditional signal processing techniques. While each reviewer has raised a number of specific concerns about the present study, there was an agreement that the following essential revisions needed to be addressed to warrant publication of the manuscript.

1. The study compares the performance of SWR detection with a CNN and with more classic signal processing methods (i.e. filters). However, several aspects of this comparison are unclear, if not misleading. First, while the CNN has obviously been optimized to detect SWRs, the study should clarify how the filter parameters were chosen. The study should demonstrate that fine-tuned filters still underperform the CNNS. Moreover, the study should present a fair comparison of the performance of each method, based on the same ground truth data, using the same number of channels, etc.

2. It is unclear what types of events are detected by the CNN and missed by the traditional approach, how can one be sure these are more physiologically relevant. Isn't it possible that some of these events result for example from an increase in spiking without clear underlying SWR? The study should provide more information regarding the nature of the events that are detected by the CNN and not the traditional approach.

3. The study should also convincingly demonstrate that the CNN can be applied to a new dataset and still outperform the spectral approach without (or at least little) re-training. This is key to validating the method.

4. The higher performance of the CNN should be demonstrated with, for example, manual scoring of false positive/negative rates. In summary, there should be no d.ubt that the CNN outperforms the spectral approach across conditions and datasets.

Please see the reviewers' comments below for more details.

*Reviewer #1 (Recommendations for the authors):*

– Several times throughout the paper the authors claim that "spectral methods fail to capture the mechanistic complexity of SWRs". I understand that they want to make the point that spectral methods are based mainly on the spectrum, and therefore constrain the variability of the events to be detected. However, this sentence is misleading as spectral methods could also be used to detect variability of events if properly tuned, even in separate runs throughout one dataset if that is what is wanted. This is a rather important point as it can be misinterpreted by the fact that those methods cannot be used to detect a variety of SWRs, which is not true if they are used properly.

– Several times the authors also claim that they are able to detect "physiologically relevant processes" with their new method, but as it is, the manuscript just shows that they can detect new events, and remain to be shown whether they are "physiologically relevant".

– In "Figure S1. Network definition and parameters" do the authors mean "Figure 1"? In Figure 1 the authors show how they tune the parameters that work best for their CNN method and from there they compare it with a filter method. Presumably, the filter method has also been passed through a parameter tuning process to be used to its best performance but this is not stated and not shown anywhere in the paper. If "relatively arbitrary" parameters are used, then this could be the explanation of why the performance of the filter method is worst compared to CNN.

– In Figure 2. What do the authors mean by absolute ground truth? I could not find a clear explanation in the text and it seems to me that the authors refer to "absolute ground truth" as the events detected by CNN? If this is the case I am not sure is the best approach to use this as a fair comparison of "absolute ground truth". Similarly, I don't think is the best approach to use the "mean reference of performance" the score of a second research (nGT) of that of the previous researcher (oGT) as the second score will inherit the fails of the first score. Instead, maybe compare the two of these metrics separately or take the average of the two of them?

– The authors should show at least one manual score of the performance of their CNN method detection, showing examples of what they might consider false positives and missed scores. In figure 2D they did it for an external dataset and they re-scored it in order to "correct" the original ground truth. They show a "false positive" that they corrected but as I understand it, if that event was not part of the ground truth is a "missed" or "false negative" event instead of a "false positive" right?

– In Figure 2E the authors show the differences between CNN with different precision and the filter method, while the performance is better the trends are extremely similar and the numbers are very close for all comparisons (except for the recall where the filter clearly performs worse than CNN) and the significance might be an effect of sample size.

– The authors claim that "some predictions not consistent with the current definition of SWR may identify different forms of population firing and oscillatory activities associated with sharp-wave", while this is true, it is the fact that by the nature of the LFP and spiking activity, typical noise of the network at low (LFP) and high (spikes) frequencies could be capture in the CNN and misinterpreted as a "relevant event".

– In Figure 5 the authors claim that they find "striking differences in firing rates and timings of SWRs detected at SO, SR and SLM", however, from the example plots in Figure 5H it is clear that except SO, all other strata follow a similar timing, with bot SO and to some extent SLM showing some misalignment in time. How confident are the authors about this variability which turn out to be significant in Figure 5H is not related to the fact that at the two sides of the dipole of the pyramidal cell layer (SO and SLM) more noise can be detected due to larger events fluctuations that not necessarily are ripple events? In other words, the events detected at SO and SLM could contain a higher percentage of false positives? Alternatively, could the variability be related to the occurrence (and detection) of similar events in neighboring spectral bands (i.e., γ events)? The authors should discuss this point in the text.

Overall, I think the method is interesting and could be very useful to detect more nuance within hippocampal LFPs and offer new insights into the underlying mechanisms of hippocampal firing and how they organize in various forms of network events related to memory. Nonetheless, I suggest clarifying the above points for better interpretability as it will also clarify the context of how the method is being validated and where it could be applied in the future.

*Reviewer #2 (Recommendations for the authors):*

The key points that are required to convincingly support the claims of the paper are:

1. Comparing the CNN to a filter approach that has access to the same number of channels and can also learn the relative weight of each.

2. Showing that the CNN significantly outperforms such a model in detecting SWRs.

3. Convincingly demonstrating that the model can be applied to new datasets with good performance and reasonable overhead.

4. Showing that the CNN can identify real, biologically relevant aspects of SWRs that a filter cannot.

---

## [Author Response]

1. The study compares the performance of SWR detection with a CNN and with more classic signal processing methods (i.e. filters). However, several aspects of this comparison are unclear, if not misleading. First, while the CNN has obviously been optimized to detect SWRs, the study should clarify how the filter parameters were chosen. The study should demonstrate that fine-tuned filters still underperform the CNNS.

See below the results of the parameter study for the filter in the very same sessions used for training the CNN. The parameters chosen (100-300Hz band, order 2) provided maximal performance in the test set. Therefore, both methods are similarly optimized along training. This is now included (page 4):

“In order to compare CNN performance against spectral methods, we implemented a Butterworth filter, which parameters were optimized using the same training set (Figure 1—figure supplement 1D).”

Moreover, the study should present a fair comparison of the performance of each method, based on the same ground truth data, using the same number of channels, etc.

Please, note that the same ground truth data was used to optimize, to test and to validate performance of both models. Regarding using the same number of channels for the offline filter detection, please see Author response image 1, a comparison of performance using different combinations of channels, from the standard detection at the SP layer (pyr) up to 4 and 8 channels (consensus detection). The filter performance is consistent across configurations and do not improve as more channels are added (actually there is a trend to decrease). We made a note in Figure 1-supp-1D, caption:

“Evaluation of the parameters of the Butterworth filter exhibiting performance F1>0.65 (green values), similar to the CNN. The chosen parameters (100-300 Hz bandwidth and order 2) are indicated by arrowheads. We found no effect of the number of channels used for the filter (1, 4 and 8 channels), and chose that with the higher ripple power”

**Author response image 1. sa2fig1:** 

2. It is unclear what types of events are detected by the CNN and missed by the traditional approach, how can one be sure these are more physiologically relevant.

We have visually validated all false positives (FP) detected by the CNN and the filter. As can be seen below the large majority of FP detected by the filter were artifacts (53.9% vs 27.7% for the CNN). The CNN detected more SW-no ripple (20.1% vs 7.8% for the filter) and events with population firing (14.9% vs 6.2%). Please, note that in many labs, detection of population firing synchrony is used as a proxy of replay events and ripples. For instance, this is exactly the issue with the Grosmark and Buzsaki 2016 ground truth, as we discuss below. We now include this analysis in the new Figure 4F. To facilitate the reader examining examples of True Positive and False Positive detections we also include a new figure (Figure 5), which comes with the executable code (see page 9)

Isn't it possible that some of these events result for example from an increase in spiking without clear underlying SWR? The study should provide more information regarding the nature of the events that are detected by the CNN and not the traditional approach.

To address this concern further, we estimated the power spectra of FP events detected by the CNN and missed by the filter and vice versa. We also considered TP events detected by both methods, as a reference. As can be seen in Author response image 2 below, for TP events detected by both methods the power spectrum displays two peaks corresponding to contributions by the sharp-wave (2-20 Hz) and the ripple (100-250 Hz) (see Oliva et al., Cell Reports 2018 for similar analysis). FP events detected exclusively by the CNN displayed a similar low frequency peak corresponding to the sharp-wave, and no dominant contribution in the high frequency band. In contrast, FP events detected exclusively by the filter more likely reflect events with higher frequency components. These FP events are among those categorized above.

Finally, we compared the features of TP events detected by both methods. There is effect of threshold in both the frequency and power values of SWR detected by the filter (upper plots). In contrast, the CNN was less sensitive, as we already discussed (bottom plots). Moreover, SWR events detected by the CNN exhibited features similar to those of the ground truth (GT), while for the filter there were some significant differences. We include this analysis in the new Figure 2 of the revised version (page 5).

3. The study should also convincingly demonstrate that the CNN can be applied to a new dataset and still outperform the spectral approach without (or at least little) re-training. This is key to validating the method.

We would like to stress that we already provided this evidence in the previous version. Please note that the CNN was applied to several new datasets without re-training: (a) 15 sessions from 5 mice never used for training and test (Figure 2A); (b) new experimental sessions for online detection (Figure 2C); (c) an external dataset (Grossmark and Buzsaki 2016) (Figure 3E); (c) new data recorded with ultra-dense Neuropixels (Figure 7). We tried to make this even clearer in the text and figures (see schemes in Figure 3D; 7A) and reinforced the message in the abstract, introduction and Discussion sections. Therefore, the manuscript already addressed this point. We apologize if that was not clear enough.

In addition, we are currently applying the CNN without re-training to data from many other labs using different electrode configurations, including tetrodes, linear silicon probes and wires. To handle with different input channel configurations, we have developed an interpolation approach, which transform data into 8-channel inputs. Results confirm very good performance of the CNN without the need for retraining. Since we cannot disclose third-party data, we have looked for a new dataset from our own lab to illustrate the case. See Author response image 3, results from 16ch silicon probes (100 μm inter-electrode separation), where the CNN performed better than the filter (F1: p=0.0169; Precision, p=0.0110; 7 sessions, from 3 mice). We found that the performance of the CNN depends on the laminar LFP profile recorded per session, as Neuropixels data illustrate. This material will be incorporated in a subsequent paper aimed to test the CNN model online using different electrode configurations and experimental preparations.

**Author response image 3. sa2fig3:** 

Last, but not least, we provide the reader with open source codes, and access to annotated data to facilitate using and disseminating the tool. In particular:– Data is deposited in the Figshare repository https://figshare.com/projects/cnn-rippledata/117897.

– The trained model is accessible at the Github repository for both Python:

https://github.com/PridaLab/cnn-ripple, and Matlab: https://github.com/PridaLab/cnnmatlab

– Code visualization and detection is shown in an interactive notebook https://colab.research.google.com/github/PridaLab/cnnripple/blob/main/src/notebooks/cnn-example.ipynb.

– The online detection Open Ephys plugin is accessible at the Github repository: https://github.com/PridaLab/CNNRippleDetectorOEPlugin – An executable figure (Figure 5) is provided:

https://colab.research.google.com/github/PridaLab/cnn-ripple-executablefigure/blob/main/cnn-ripple-false-positive-examples.ipynb

4. The higher performance of the CNN should be demonstrated with, for example, manual scoring of false positive/negative rates. In summary, there should be no doubt that the CNN outperforms the spectral approach across conditions and datasets.

We have manually scored the FP of both the filter and the CNN, as shown before. The FN rate is 1-R, where R is recall, and it is very similar with a bit lower value for the CNN (CNN 0.34; filter 0.38). The FP rate is defined as FP/N, where N are all events in the ground truth that are not SWR. Since SWR are very erratic and most windows do not contain events, the resulting FP rate was very close to 0 for both methods. For this reason, we relied in estimating P (precision), R (recall) and F1, which are higher for the CNN (Figure 2A). We feel there is no doubt that the CNN outperformed the spectral filter across conditions and datasets.

Please see the reviewers' comments below for more details.

Please, see our responses to reviewers’ comments

Reviewer #1 (Recommendations for the authors):– In "Figure S1. Network definition and parameters" do the authors mean "Figure 1"? In Figure 1 the authors show how they tune the parameters that work best for their CNN method and from there they compare it with a filter method. Presumably, the filter method has also been passed through a parameter tuning process to be used to its best performance but this is not stated and not shown anywhere in the paper. If "relatively arbitrary" parameters are used, then this could be the explanation of why the performance of the filter method is worst compared to CNN.

This is now addressed in the new Figure 1-supp-1. The parameters chosen (100-300Hz band, order 2) provided maximal performance in the test set. Therefore, both methods are similarly optimized along training. This is now included (page 4)

– In Figure 2. What do the authors mean by absolute ground truth? I could not find a clear explanation in the text and it seems to me that the authors refer to "absolute ground truth" as the events detected by CNN? If this is the case I am not sure is the best approach to use this as a fair comparison of "absolute ground truth". Similarly, I don't think is the best approach to use the "mean reference of performance" the score of a second research (nGT) of that of the previous researcher (oGT) as the second score will inherit the fails of the first score. Instead, maybe compare the two of these metrics separately or take the average of the two of them?

We thank the reviewer for this comment. Just to clarify, by the absolute ground truth we meant the ‘whole truth’ that includes all SWR (which is unknown). Since the experts’ ground truths are similar at about 70%, there are non-tagged SWR missed in the individual ground truths. Thus, as we increase the number of experts we should converge on the absolute ground truth because they are adding more events to the pool. The analysis is about the effect of the experts’ ground truth on performance. What we show is that when we consolidated the two GTs, meaning those events detected by different experts add together, the CNN improved performance but not the filter. This makes sense because this rescues some of those events that could be arguable whether they reflect SWR or not to the eyes of individual experts which are more typically missed by the filter (e.g. population synchrony with sharp-wave or sharp-wave no ripples). Please, note that the two experts annotated data independently, but the GT by the new expert is used for validation purposes only (no training).

To avoid misunderstanding we removed the reference to the absolute ground truth from Figure 3B, and clarified the issue:

“To evaluate the impact of these potential biases, we used the ground truth from a second expert in the lab for validation purposes only (3403 events, n=14 sessions, 7 mice).” (page 6) “In contrast, the filter failed to exhibit such an improvement, and performed worse when tested against the consolidated ground truth (one-way ANOVA for models, F(2)=0.02, p=0.033) (Figure 3B). “(page 7). “CNN performance improves when confronted with the consolidated ground truth, supporting that shared community tagging may help to advance our understanding of SWR definition” (page 8)

– The authors should show at least one manual score of the performance of their CNN method detection, showing examples of what they might consider false positives and missed scores. In figure 2D they did it for an external dataset and they re-scored it in order to "correct" the original ground truth. They show a "false positive" that they corrected but as I understand it, if that event was not part of the ground truth is a "missed" or "false negative" event instead of a "false positive" right?

We apologize for confusion. Note we manually scored all False Positive events from the training and validation dataset (17 sessions, from 7 mice). This is now shown for both the filter and the CNN in Figure 4F and examples are shown in the executable Figure 5. Regarding the new Figure 3D,E, we reevaluated events from the external data set. Please, note that we chose the Grosmark and Buzsaki 2016 dataset because SWR detection was conditioned on the coincidence of both population synchrony and LFP definition thus providing a “partial ground truth” (i.e. SWR without population firing were not annotated in the dataset). Thus, we revalidated False Positive detection. This is a perfect example of how the experimental goal (examining replay and thus relying in population firing plus LFP definitions) limits the ground truth. We have clarified the text.

“To evaluate this point further, and to test the capability of the CNN to generalize beyond its original training using head-fixed mice data, we used an externally annotated dataset of SWR recorded with high-density silicon probes from freely moving rats (Grosmark and Buzsáki, 2016) (Figure 3D; 2041 events; 5 sessions from 2 rats; Sup.Table.1). In that work, SWR detection was conditioned on the coincidence of both population synchrony and LFP definition, thus providing a “partial ground truth” (i.e. SWR without population firing were not annotated in the dataset).” See page 7.

– In Figure 2E the authors show the differences between CNN with different precision and the filter method, while the performance is better the trends are extremely similar and the numbers are very close for all comparisons (except for the recall where the filter clearly performs worse than CNN) and the significance might be an effect of sample size.

This refers again to the new Figure 3D,E of the external dataset. Following the advice we have added more samples to improve statistical testing (n=5 sessions from 2 rats). We now report significant differences. See Figure 3E.

– The authors claim that "some predictions not consistent with the current definition of SWR may identify different forms of population firing and oscillatory activities associated with sharp-wave", while this is true, it is the fact that by the nature of the LFP and spiking activity, typical noise of the network at low (LFP) and high (spikes) frequencies could be capture in the CNN and misinterpreted as a "relevant event".

As suggested, we have validated al FP predictions (new Figure 4F; Figure 5). We also evaluate the quantitative features of SWR events detected by the filter and the CNN, and compare them with the GT (Figure 2B). Finally, we discuss this point in the revised version. In particular:

“While we cannot discard noisy detection from a continuum of LFP activity, our categorization suggest they may reflect processes underlying buildup of population events (de la Prida et al., 2006). In addition, the ability of CA3 inputs to bring about γ oscillations and multi-unit firing associated with sharp-waves is already recognized (Sullivan et al., 2011), and variability of the ripple power can be related with different cortical subnetworks (Abadchi et al., 2020; RamirezVillegas et al., 2015).” (page 16).

– In Figure 5 the authors claim that they find "striking differences in firing rates and timings of SWRs detected at SO, SR and SLM", however, from the example plots in Figure 5H it is clear that except SO, all other strata follow a similar timing, with bot SO and to some extent SLM showing some misalignment in time.

We apologize for generating confusion. We meant that the analysis was performed by comparing properties of SWR detected at SO, SR and SLM using z- values scored by SWR detected at SP only. We clarified this point in the revised version (page 14).

How confident are the authors about this variability which turn out to be significant in Figure 5H is not related to the fact that at the two sides of the dipole of the pyramidal cell layer (SO and SLM) more noise can be detected due to larger events fluctuations that not necessarily are ripple events? In other words, the events detected at SO and SLM could contain a higher percentage of false positives? Alternatively, could the variability be related to the occurrence (and detection) of similar events in neighboring spectral bands (i.e., γ events)? The authors should discuss this point in the text.

As we showed above there is no differences of False Positive detections between SO, SR and SLM layers. Regarding the potential effect of background activity, we now discuss this point:

“While we cannot discard noisy detection from a continuum of LFP activity, our categorization suggest they may reflect processes underlying buildup of population events (de la Prida et al., 2006). In addition, the ability of CA3 inputs to bring about γ oscillations and multi-unit firing associated with sharp-waves is already recognized (Sullivan et al., 2011), and variability of the ripple power can be related with different cortical subnetworks (Abadchi et al., 2020; RamirezVillegas et al., 2015).” (Page 16)

Overall, I think the method is interesting and could be very useful to detect more nuance within hippocampal LFPs and offer new insights into the underlying mechanisms of hippocampal firing and how they organize in various forms of network events related to memory. Nonetheless, I suggest clarifying the above points for better interpretability as it will also clarify the context of how the method is being validated and where it could be applied in the future.

Thank you for constructive comments

Reviewer #2 (Recommendations for the authors):The key points that are required to convincingly support the claims of the paper are:

We thank the reviewer for providing this checklist. Please, see below.

1. Comparing the CNN to a filter approach that has access to the same number of channels and can also learn the relative weight of each.

We have examined the filter when applied to the same number of channels (8-channels), which did not improve performance. We also optimized the filter parameters using the 2 training sessions, similar to the CNN. Our data still support the CNN detection capability better than the filter. Please, note that training an algorithm to learn the weight of each filter it will be an ANN itself and this is an entirely new project.

2. Showing that the CNN significantly outperforms such a model in detecting SWRs.

Here, we have provided evidence supporting that the CNN significantly outperforms the filter using: (a) offline validation in datasets not used for training (mice recorded head-fixed); (b) online validation in new experiments; (c) an independent dataset tagged by two experts; (d) an external data set from different preparation and species (freely moving rats) to illustrate the effect of the definition of the ground truth (SWR without population firing were not originally annotated in the dataset); (e) new dataset of Neuropixels recordings from head-fixed mice. In addition, in response to the reviewers we showed: (a) that using 8-channels and a consensus optimization did not increased performance of the filter; (b) preliminary results from coarse-density 16ch recorded (100 µm inter-elecrode distance), analyzed by the CNN using interpolation strategies, confirmed the ability of the trained model to generalize.

3. Convincingly demonstrating that the model can be applied to new datasets with good performance and reasonable overhead.

Please, note that in the original version we do apply the model to different new datasets without raining. See response above (point 2).

4. Showing that the CNN can identify real, biologically relevant aspects of SWRs that a filter cannot.

We have validated all detections by both the filter and the CNN to show that the network clearly identify more biologically relevant events such as SW-no ripple and SWR with population firing. Also, we provide comparisons of the features of TP events detected by both methods, to support that SWR detected by the CNN exhibited features more similar to those of the ground truth than those detected by the filter.